# Ventromedial prefrontal cortex compression during concept learning

Michael L. Mack [1]*, Alison R. Preston [2,3,4,7] & Bradley C. Love [5,6,7]

Prefrontal cortex (PFC) is thought to support the ability to focus on goal-relevant information by filtering out irrelevant information, a process akin to dimensionality reduction. Here, we test this dimensionality reduction hypothesis by relating a data-driven approach to characterizing the complexity of neural representation with a theoretically-supported computational model of learning. We find evidence of goal-directed dimensionality reduction within human ventromedial PFC during learning. Importantly, by using computational predictions of each participant's attentional strategies during learning, we find that that the degree of neural compression predicts an individual's ability to selectively attend to concept-specific information. These findings suggest a domain-general mechanism of learning through compression in ventromedial PFC.

[1] Department of Psychology, University of Toronto, Toronto, ON, Canada. [2] Department of Psychology, The University of Texas at Austin, Austin, TX, USA. [3] Center for Learning and Memory, The University of Texas at Austin, Austin, TX, USA. [4] Department of Neuroscience, The University of Texas at Austin, Austin, TX, USA. [5] Experimental Psychology, University College London, London, UK. [6] Alan Turing Institute, London, UK. [7] These authors contributed equally: Alison R. Preston, Bradley C. Love *email: mack.michael@gmail.com

Prefrontal cortex (PFC) is sensitive to the complexity of incoming information[1] and theoretical perspectives suggest that a core function of PFC is to focus representation on goal-relevant features by filtering out irrelevant content[2,3]. In particular, ventromedial PFC (vmPFC) is thought to represents the latent structures of experience[4,5], coding for causal links[6], and task-related cognitive maps[7]. At the heart of these accounts is the hypothesis that during learning vmPFC may perform data reduction on incoming information, compressing task-irrelevant features, and emphasizing goal-relevant information structures. This compression process is goal-directed and akin to how attention in category learning models dynamically selects features that have proven predictive across recent learning trials[8,9]. Although emerging evidence suggests structured representations occur in the rodent homolog of vmPFC[10,11], such coding in human vmPFC remains poorly understood. Here, we directly assess the data reduction hypothesis by leveraging an information-theoretic approach in human neuroimaging to measure how goal-driven learning is supported by attention updating processes in vmPFC.

We focused on concept learning, given the recent findings that vmPFC represents conceptual information in an organized fashion[12,13]. Participants learned to classify the same insect images (Fig. 1a), composed of three features that could take on two values (thick/thin legs, thick/thin antennae, pincer/shovel mandible), across three different learning problems[14]. These learning problems were defined by rules that required consideration of different numbers of features to successfully classify (see Table 1): the low category complexity problem was unidimensional (e.g., insects living in warm climates have thick legs,

cold climate insects have thin legs), the medium category complexity problem depended on two features (e.g., insects from rural environments have thick antennae and shovel mandible or thin antennae and pincer mandible, urban insects have thick antennae and pincer mandible or thin antennae and shovel mandible), and the high category complexity problem required all three features (i.e., each insect's class was uniquely defined by a combination of features). By using the same stimuli for all three problems, the manipulation of conceptual complexity allowed us to target goal-specific learning processes.

This design allows us to directly test whether compression of neural representations corresponds with the complexity of the problem-specific conceptual structure during learning. Complexity and compression have an inverse relationship; the lower the complexity of a conceptual space, the higher the degree of compression. For instance, in learning the unidimensional problem, variance along the two irrelevant feature dimensions can be compressed resulting in a lower complexity conceptual space. In contrast, learning the high complexity problem would result in less compression because all three feature dimensions must be represented, resulting in a more complex conceptual space relative to the unidimensional problem. Differences in complexity across the three learning problems thus provide a means for testing how learning shapes the dimensionality of neural concept representations. Namely, brain regions involved in goal-directed data compression should learn by building internal models that adapt to the complexity of the problems in order to represent information relevant to the task at hand.

To test this prediction, we recorded functional magnetic resonance imaging (fMRI) data while participants learned the three problems and measured the degree that multivoxel activation patterns were compressed through learning using principal component analysis (PCA), a method for low-rank approximation of multidimensional data[15]. We demonstrate that neural codes in vmPFC distinctly reflect the conceptual complexity of the learning problems and that this neural signature of dimensionality reduction corresponds with participants' ability to learn goal-relevant information.

## Results

**Learning-related neural compression.** Trial-level whole-brain activation patterns for each insect image were estimated using the least squares-separate approach[16]. These trial-specific activation patterns were then submitted to principal component analyses (PCA) and the number of principal components (PCs) that were necessary to explain 90% of the variance across trials within a learning block was used to calculate an index of neural

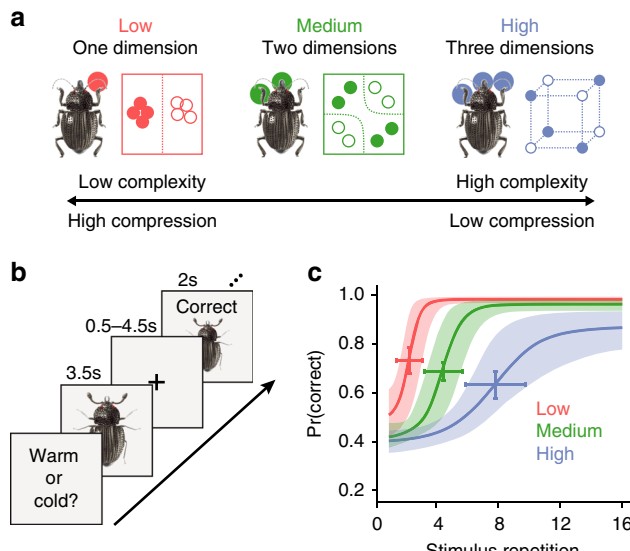

**Fig. 1 Experimental schematic and behavioral results (N = 23). a** The learning problems differed in rule complexity (see Supplementary Table 1). The low complexity problem was unidimensional (e.g., antennae size), medium complexity required a conjunction of two features (e.g., leg size and mandible shape), and high complexity required all three features. **b** Learning trials consisted of presentation of a stimulus for 3.5 s, followed by a fixation cross for 0.5–4.5 s, and then a feedback display for 2 s that included the stimulus, accuracy of the response, and the correct category. Learning trials were separated by a delay of 2–6 s of fixation. **c** The probability of a correct response increased across stimulus repetitions. The rate of learning differed according to the complexity of the problems. Bands represent 95% confidence intervals of the logistic regression model and error bars represent 95% prediction intervals for the midpoint of accuracy across participants.

**Table 1 Stimulus features and class associations for the three learning problems.**

| Stimulus | Feature attribute | | | Problem complexity | | |
|---|---|---|---|---|---|---|
| | 1 | 2 | 3 | Low | Medium | High |
| 1 | 0 | 0 | 0 | A | A | B |
| 2 | 0 | 0 | 1 | A | B | A |
| 3 | 0 | 1 | 0 | A | B | A |
| 4 | 0 | 1 | 1 | A | A | B |
| 5 | 1 | 0 | 0 | B | A | A |
| 6 | 1 | 0 | 1 | B | B | B |
| 7 | 1 | 1 | 0 | B | B | B |
| 8 | 1 | 1 | 1 | B | A | A |

Each of the eight stimuli is represented by the binary values of the three feature attributes. The stimuli are assigned to different classes (A or B) across the low, medium, and high complexity learning problems according to rules that depend on one, two, or three of the feature attributes, respectively

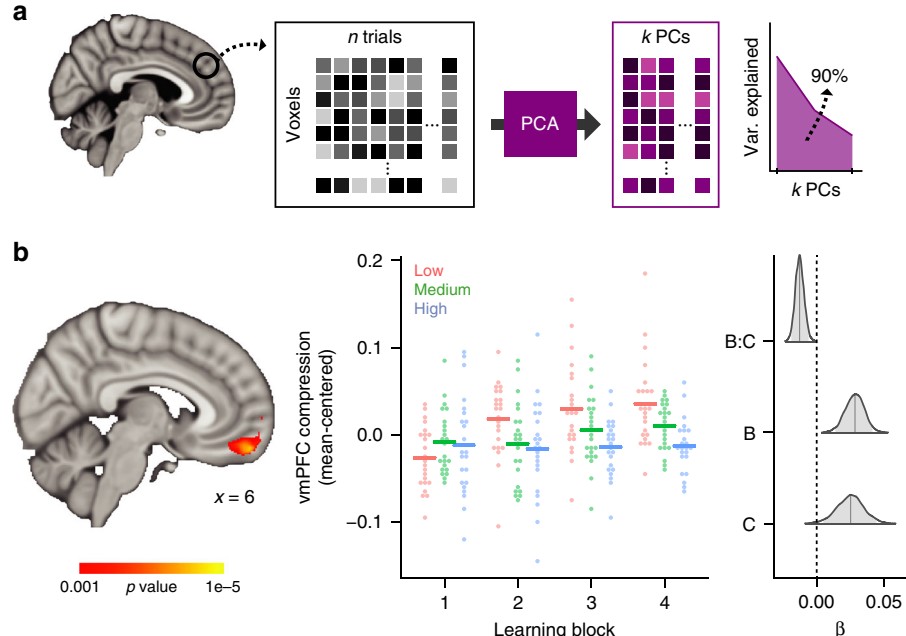

**Fig. 2 Neural compression analysis schematic and results ($N = 23$). a** Principal component analysis (PCA) was performed on neural patterns evoked for each of $n$ trials within a learning block. The number of principal components (PC) required to explain 90% of the variance ($k$) was used to calculate a neural compression score (1-$k/n$). We quantified neural compression as a function of problem complexity and learning block; the interaction of these factors reflects changes in the complexity of neural representations that emerge with learning. **b** A whole-brain voxel-wise linear mixed effects regression revealed a vmPFC region that showed a significant interaction between learning block and problem complexity. See Supplementary Figure 1 for main effect maps of learning block and problem complexity. The nature of the interaction in the vmPFC region is depicted in the middle panel; points represent compression at the cluster's peak voxel for each participant and the horizontal lines depict the group average. The right graph plots the results of a Bayesian-estimated linear mixed effects regression of neural compression from the peak voxel of the vmPFC cluster. Posterior distributions of coefficients from the regression model are depicted for the factors of learning block (**b**), complexity (**c**), and their interaction (**b:c**). Shaded regions within the distributions represent 95% high-density intervals. These data and regression results are displayed only to demonstrate the nature of the interaction effect in the vmPFC cluster and do not represent an independent statistical analysis.

compression (i.e., fewer PCs reflects more neural compression; Fig. 2a). This measure of neural compression was calculated across the whole brain with searchlight methods[17] for each learning block in each problem. We then identified brain regions that reduce dimensionality with learning (i.e., learn to represent the less complex problems with fewer dimensions) by conducting a voxel-wise linear mixed effects regression on the searchlight compression maps. Specifically, at each voxel, we assessed how neural compression changed as a function of learning block and problem complexity and their interaction.

Throughout the entire brain, only a region within vmPFC showed an interaction of problem complexity and learning block (peak coordinates [4, 54, −18]; 653 voxels; voxel-wise threshold = 0.001, cluster extent threshold = 0.05; Fig. 2b; see Supplementary Fig. 1 and Supplementary Table 1 for main effects of problem complexity and learning block). The nature of the interaction within this cluster was assessed with a Bayesian-estimated mixed effects linear regression and showed that vmPFC compression corresponded with problem complexity and emerged over learning blocks (peak: $\beta_{\text{mean}} = -0.013$, 95% HDI = [−0.019, −0.006], $P < 0.001$). Importantly, the interaction effect was independent of problem order, individual differences in learning performance, and remained when looking at only the low and medium complexity problems (see "Methods" section for details about the voxel-wise regression modeling and control analyses). Moreover, there was no interaction of learning block and problem complexity when examining univariate vmPFC activation (Bayesian-estimated mixed effects linear regression: $\beta_{\text{mean}} = -1.309$, 95% HDI = [−82.7, 126.7], $P = 0.93$), ruling out an explanation based on problem difficulty impacting overall

neural activation. Because the stimuli were identical across the three problems, this finding demonstrates that learning-related compression is goal-specific, with vmPFC requiring fewer dimensions for less complex goals.

**Neural compression relates to attentional strategies**. To evaluate whether vmPFC compression tracked changes in attentional allocation, we characterized the participant-specific attentional weights given to each stimulus feature across the three problems using a computational learning model[8]. Attention weight compression indexed changes in attentional allocation based on model fits to behavior; low attention compression indicates equivalent weighting to all three features, whereas high attention compression indicates attention directed to only one feature[18]. We found that attention compression varied with the interaction of learning block and conceptual complexity (Bayesian-estimated mixed effects linear regression: $\beta_{\text{mean}} = -0.028$, 95% HDI = [−0.035, −0.020], $P < 0.001$). That vmPFC neural compression and model-based attention compression demonstrated similar relationships with conceptual complexity suggests a potential link between the behavioral/model and neural signatures of dimensionality reduction.

To assess this relationship, we evaluated whether the compression of participants' attention weights was predicted by vmPFC neural compression at the individual participant level. Specifically, if the ability to compress neural representations in a problem-appropriate fashion is related to participants' ability to attend to problem-relevant features, the prediction follows that participants with more neural compression for a given problem will also show more selective attention, thus higher attention

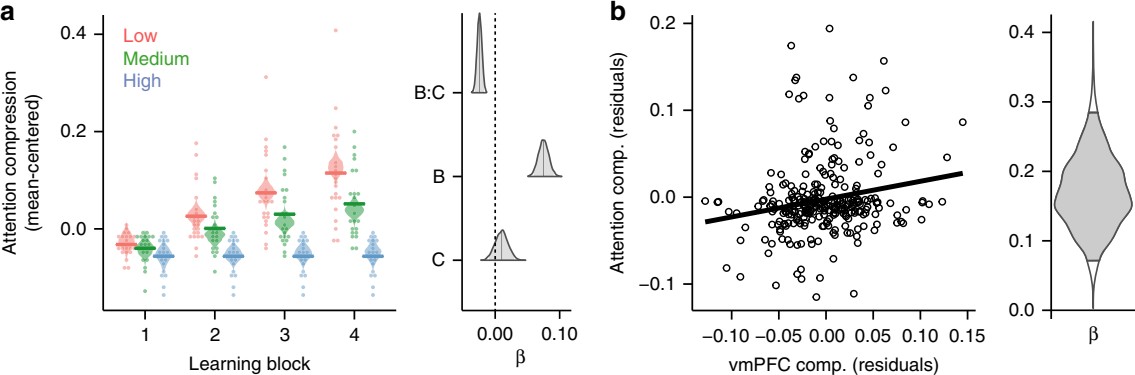

**Fig. 3 Relationship between model-based attention weighting and vmPFC neural compression (N = 23). a** Attention compression (i.e., degree of attention selectivity as derived from cognitive model fits to behavior) emerged according to feature relevancy across the problems with highest compression for low complexity followed by medium and high complexity by the end of learning. Points depict individual participants, horizontal lines are group averages, and violin plots depict the distribution of posterior predictions from the Bayesian-estimated linear regression. The right panel depicts posterior distributions of regression coefficients for learning block (**b**), complexity (**c**), and their interaction (**b:c**). The shaded region within the distributions marks the 95% high-density interval. **b** vmPFC neural compression predicted the degree of problem-specific computational model-based attention compression across learning blocks and problems. Both vmPFC neural compression and model-based attention compression are plotted as partial residuals of separate regression models that regress out factors of learning block, problem complexity, accuracy, and learning order. The solid line depicts the best-fitting regression line of the partial residuals. The violin plot depicts the posterior distribution of regression coefficient relating neural compression and attention compression. The shaded area bounded by black lines within the distribution mark the 95% high-density interval.

compression values. This hypothesis was confirmed with a Bayesian-estimated mixed effects regression analysis that took into account differences across learning block, problem complexity, accuracy, and learning order ($\beta_{mean} = 0.168$, 95% HDI = [0.072, 0.277], $P = 0.0005$; see Fig. 3b). Importantly, this relationship remained when restricting analysis to the low and medium complexity problems, which were counterbalanced for learning order (Bayesian-estimated mixed effects linear regression: $\beta_{mean} = 0.210$, 95% HDI = [−0.082, 0.368], $P = 0.0005$). Thus, even after controlling for differences in neural and attention compression due to learning block and problem complexity, the degree of problem-specific compression in vmPFC representations significantly predicted participants' attentional strategies throughout learning.

**Category coding in neural compression**. Although vmPFC neural compression tracks model-based predictions of learning, this link between learning problem-specific coding and neural representation is ultimately indirect. The neural compression findings may be due to learning-related changes in neural representation that highlight within-category similarities and differentiate between-category differences or due to other factors unrelated to the category structure of the problem at hand. To directly assess the degree of category coding present in vmPFC compression, we analyzed how trials loaded onto the PCs. Specifically, we hypothesized that if neural compression is driven by category-specific coding in activation patterns, trials will load on the PCs similarly within category, but differently between category (Fig. 4a). In contrast, if neural compression is due to factors not related to category representation, trials from both categories will load similarly on the PCs. We found support for the former (Fig. 4b and Table 3) such that category discrimination in PCA loadings increased over learning blocks (Bayesian-estimated mixed effects linear regression: $\beta_{mean} = 0.008$, 95% HDI = [0.004, 0.011], $P < 0.001$) and was highest for the low followed by medium and high complexity problems ($\beta_{mean} = −0.003$, 95% HDI = [−0.0048, −0.0013], $P < 0.001$). Thus, by directly assessing the structure of the PCA results, we find that vmPFC compression is driven by activation patterns that discriminate categories based on current task goals.

## Discussion

By focusing on a mechanism by which vmPFC may form and represent concepts through goal-sensitive dimensionality reduction, we show that neural representations in a vmPFC subregion are shaped by experience. And, this shaping is adaptive, promoting efficient representation of information that focuses on encoding features that are most predictive of positive outcomes for a given goal. Importantly, by evaluating behavior through the lens of a theoretically oriented computational model, we demonstrate that compression in vmPFC unfolds over the course of learning in a manner consistent with the learning mechanisms of SUSTAIN[8,9]. These findings provide a quantitative account of vmPFC's potential role in the coding of efficient schematic models or cognitive maps[7,12,13,19], specifically in the conceptual domain.

Successfully learning new concepts requires attending to goal-diagnostic features and ignoring irrelevant information to build abstract representations that capture the structure defining a concept[8]. Viewed in these terms, concept learning has many parallels to schema formation, a vmPFC-related function first identified in lesion studies in the memory literature[20]. Schemas are defined as structured memory networks that represent associative relationships among prior experiences and provide predictions for new experiences[4,5,21,22]. Schema-related memory behaviors are significantly impacted by vmPFC lesions. For example, vmPFC lesion patients exhibit a reduced influence of prior knowledge during recognition of items presented in schematically congruent contexts compared with healthy controls[23]. Moreover, vmPFC lesions have been associated with a marked inability to differentiate schema-related concepts from concepts inappropriate for a given schema[24]. From this work, it is clear that vmPFC is necessary for retrieving generalized representations built from prior events that are relevant to current experience. Such guided retrieval of relevant learned representations is a key to building new concepts.

A key proposal of the SUSTAIN computational model we leveraged is that concept learning is decidedly goal-based, with concept representations adaptively formed to reflect the task at hand[8]. Recent rodent and human work support this proposal with findings that vmPFC representations are goal-specific in

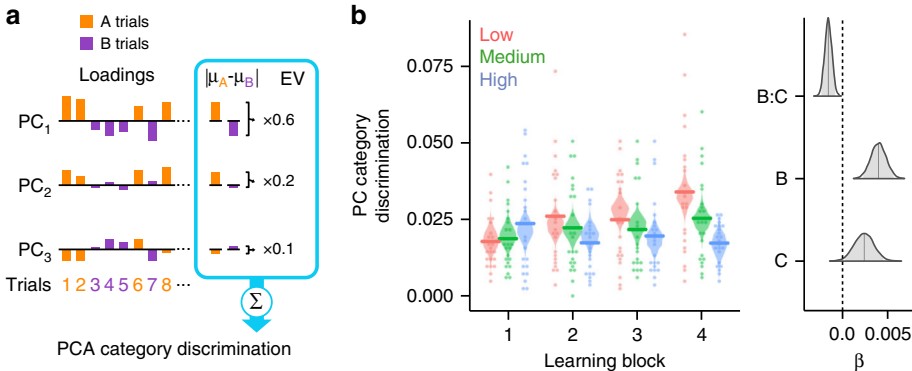

**Fig. 4 Category discrimination in neural compression ($N = 23$). a** Category discrimination in PCA neural compression was indexed by the difference in loadings for the two categories. The absolute difference between the average loadings for category A trials (orange) and B trials (purple) for each PC were weighted by the PC's explained variance ratio and summed. **b** Category discrimination increased across learning blocks with the highest discrimination for low followed by medium and high problem complexity. Points depict individual participants, horizontal lines are group averages, and violin plots depict the distribution of posterior predictions from the Bayesian-estimated linear regression. The right panel depicts posterior distributions of regression coefficients for learning block (**b**), complexity (**c**), and their interaction (**b:c**). The shaded region within the distributions marks the 95% high-density interval.

nature, at least at the end of learning. Specifically, neural ensembles in the rodent homolog of vmPFC have been demonstrated to represent higher order goal states that relate stimuli to behaviorally relevant value[10,11,25]. Similarly, one human neuroimaging study recently localized representations of a complex task space relating 16 different task states to vmPFC activation patterns[7]. Importantly, these vmPFC representations of task states predicted participants' behavioral performance. It is worth noting that much of the human task/goal-state literature refers to functions of orbitofrontal cortex (OFC). The specific boundaries of vmPFC and its relation to OFC are muddied by definitions arising from anatomical and/or functional distinctions made across somewhat separate literatures; however, there is a general consensus that vmPFC largely overlaps with the medial portion of OFC[26]. Future work will continue to uncover the spatial organization of functional aspects of OFC and vmPFC, but the existing literature does support the notion that vmPFC organizes knowledge based on goals to promote flexible behaviors.

Our findings provide important evidence for the role of vmPFC during the formation of conceptual maps of experience. Although theoretical perspectives highlight the importance of vmPFC in cognitive map formation[2,19], empirical work has failed to directly examine the computations of vmPFC contributions during encoding. Instead, evidence is limited to representations that are established after long periods of training[7,12]. Relatedly, most current models of vmPFC function in memory focus on its role in biasing reactivation of relevant prior experiences via the hippocampus[27]. Few studies target the role of vmPFC during encoding. We know that vmPFC interaction with memory centers during encoding[4,22,28,29], but we do not know how vmPFC knowledge representations emerge and support behavior. Our findings provide evidence for vmPFC potentially playing a role in encoding processes that build goal-specific mental models. The current neural findings are ambiguous as to whether vmPFC is directly implicated in forming conceptual representations or simply reflects representations learned elsewhere. For example, rodent models have implicated similar dimensionality-reduction processes to the basal ganglia with outputs influencing frontal coding[30]. However, by linking vmPFC coding to the learning mechanisms defined in SUSTAIN, our results suggest that vmPFC may influence encoding through dimensionality reduction wherein selective attention highlights goal-specific information and discards irrelevant dimensions. That vmPFC was the only region identified in our analysis provides more support for

such a direct influence at encoding: inputs to vmPFC are weighted to select goal-related information and discard irrelevant features in order to efficiently map input to a goal-directed action. This efficient mapping may then be fed back to memory centers (i.e., hippocampus) and even high-level vision areas[31] to impact neural coding of learning experiences[28]. This theorized role for vmPFC coding during learning offers a strong hypothesis for future work investigating flexible goal-oriented behavior.

Our hypothesized view of vmPFC function is based on SUS-TAIN's formalism of highly interactive mechanisms of selective attention and learning[8], functions theoretically mapped onto interactions between PFC and the hippocampus[9,28,32]. Support for this view is found in recent patient work that has demonstrated a causal link between attentional processes and vmPFC function in decision making[33–35]. These studies have shown that lesions to vmPFC disrupt attentional guidance based on prior experience with cue-reward associations[35], learning the value of task-diagnostic features during probabilistic learning[33], and value comparison during reinforcement learning[34]. These findings have been recently extended to healthy humans in a neuroimaging study which demonstrated that value signals in vmPFC are dynamically biased by attention during reinforcement learning[13]. Relatedly, recent rodent work demonstrates the bidirectional flow of information between the rodent homolog of vmPFC and hippocampus during context-guided memory encoding and retrieval[36–38]. Coupled with the recent demonstration of hippocampal–vmPFC functional coupling during concept learning[28], the current findings align well with the view that vmPFC is critical for evaluating and representing information in learning and decision making.

One limitation of the current work is that we evaluated our neural compression findings in light of only one computational model. Although SUSTAIN is a well-established model that explains many learning behaviors[8], is theoretically motivated by neural mechanisms of learning[9,32], and has been successfully linked to neural measures of concept learning[28,39,40], an alternative model with similar predictions for attentional tuning over the course of learning may also account for our neural compression findings. Future studies could leverage our data-driven measure of neural dimensionality reduction as a target index of learning for adjudicating between formal cognitive models.

The neural compression method proposed here offers a unique approach for evaluating the informational value of neural representations. One limitation to this approach, however, is the

need for stable trial- or condition-level general linear model (GLM) estimates of BOLD signal. Such univariate estimates in brain regions with lower signal-to-noise ratios may be noisy which would bias PCA towards less dimensionality reduction. This limitation is true of any analysis that depends on single-trial GLM estimates, including many forms of representational similarity analysis[41] and beta-series connectivity methods[42]. In the current study, only within-participant factors were considered, thus observing a significant effect requires a relative difference in neural compression across learning blocks and problem complexity within a specific brain region. As such, lower signal-to-noise ratios within a specific brain region may have influenced trial-level GLM estimates but would have at worst led to missed effects, not false positives. Importantly, the category discrimination present in the PCA loadings for the vmPFC region (Fig. 4) suggest that the vmPFC neural compression findings are related to problem-specific dimensionality reduction rather than simple changes in BOLD signal-to-noise ratios.

The current neural compression findings are based on a data-driven approach, but their link to mechanistic changes in SUSTAIN's account of learning behavior provide a useful theoretical interpretation. Such an approach actually provides an answer to questions of circularity in linking brain measures to behavior[43]. For example, in the current study, the structure of neural representations across learning was quantified with a data-driven PCA method that was ignorant to experimental conditions. The vmPFC region showing neural compression effects was identified by an interaction of learning block and problem complexity with no regard for the direction of the effects. Separately, participant behavior was characterized with SUSTAIN to derive a measure of attention compression. Thus, measures of brain representation and voxel selection criteria were independently quantified from the model-based predictions of learning behavior.

Although combining data-driven methods to understanding brain dynamics with formal psychological theories is a fruitful approach[44–46], temporal and representational scales of data analytics and theory may not align. In the current work, the neural compression measure is necessarily computed over a block of several trials, whereas SUSTAIN's predictions of learning unfold trial-by-trial. Also, PCA-based neural compression captures the complexity of neural representations based on the inherent structure of the data rather than model-based predictions[28]. Thus, neural compression offers a coarser characterization of neural dynamics. We anticipate that future extensions of neural compression methods will allow for characterizations of representational complexity at finer scales and tighter synchrony with formal theories.

In summary, we show that learning can be viewed as a process of goal-directed dimensionality reduction and that such a mechanism is apparent in vmPFC neural representations throughout learning. Thus, we suggest that vmPFC may play a critical role not only in representing conceptual content, but in the process of learning concepts. Notably, dimensionality reduction through selective attention offers a reconciling account of many processes associated with vmPFC including schema representation[47], latent casual models[7], grid-like conceptual maps[12], and value coding[48,49].

## Methods

**Participants**. Twenty-three volunteers (11 females, mean age 22.3 years old, ranging from 18 to 31 years) participated in the experiment. All subjects were right handed, had normal or corrected-to-normal vision, and were compensated $75 for participating.

The methods used in the current study are novel; however, related category learning experiments have been employed in several previous studies that focus on analyses of fMRI activation patterns[44,50]. Given the sample sizes in these studies

($N = 20, 22, 22$), as well as our previous experience with functional imaging of the whole brain, we set a target minimum samples size of 20 participants.

**Stimuli**. Eight color images of insects were used in the experiment (Fig. 1a). The insect images consisted of one body with different combinations of three features: legs, mouth, and antennae. There were two versions of each feature (thick or thin legs, shovel or pincer mandible, and thick or thin legs). The eight insect images included all combination of the three features. The stimuli were sized to $300 \times 300$ pixels.

**Procedures for the learning problems**. After an initial screening and consent in accordance with the University of Texas Institutional Review Board, participants were instructed on the classification learning problems. Participants then performed the problems in the MRI scanner by viewing visual stimuli back-projected onto a screen through a mirror attached onto the head coil. Foam pads were used to minimize head motion. Stimulus presentation and timing was performed using custom scripts written in Matlab (Mathworks) and Psychtoolbox (https://www.psychtoolbox.org) on an Apple Mac Pro computer running OS X 10.7.

Participants were instructed to learn to classify the insects based on the combination of the insects' features using the feedback displayed on each trial. As part of the initial instructions, participants were made aware of the three features and the two different values of each feature. Before beginning each classification problem, additional instructions that described the cover story for the current problem and which buttons to press for the two insect classes were presented to the participants. One example of this instruction text is as follows: "Each insect prefers either Warm or Cold temperatures. The temperature that each insect prefers depends on one or more of its features. On each trial, you will be shown an insect and you will make a response as to that insect's preferred temperature. Press the '1' button under your index finger for Warm temperatures or the '2' button under your middle finger for Cold temperatures". The other two cover stories involved classifying insects into those that live in the Eastern vs. Western hemisphere and those that live in an Urban vs. Rural environment. The cover stories were randomly paired with the three learning problems for each participant. After the instruction screen, the four fMRI scanning runs (described below) for that problem commenced, with no further problem instructions. After the four scanning runs for a problem finished, the next problem began with the corresponding cover story description. Importantly, the rules that defined the classification problems were not included in any of the instructions; rather, participants had to learn these rules through trial and error.

The three problems, the participants learned, were structured such that perfect performance required attending to a distinct set of feature attributes (Fig. 1a). For the low complexity problem, class associations were defined by a rule depending on the value of one feature attribute. For the medium complexity problem, class associations were defined by an XOR logical rule that depended on the value of the two feature attributes that were not relevant in the low complexity problem. For the high complexity problem, class associations were defined such that all feature attributes had to be attended to respond correctly. As such, different features were relevant for the three problems and successful learning required a shift in attending to and representing those feature attributes most relevant for the current problem. Critically, by varying the number of diagnostic feature attributes across the three problems, the representational space for each problem had a distinct informational complexity.

The binary values of the eight insect stimuli along with the class association for the three learning problems are depicted in Table 1. The stimulus features were randomly mapped onto the attributes for each participant. These feature-to-attribute mappings were fixed across the different classification learning problems within a participant. After the high complexity problem, participants learned the low and medium problems in sequential order. The learning order of the low and medium problems was counterbalanced across participants. This problem order was used for purposes described in a prior analysis of this data[28].

The classification problems consisted of learning trials (Fig. 1a) during which an insect image was presented for 3.5 s. During stimulus presentation, participants were instructed to respond to the insect's class by pressing one of two buttons on an fMRI-compatible button box. Insect images subtended $7.3° \times 7.3°$ of visual space. The stimulus presentation period was followed by a 0.5–4.5 s fixation. A feedback screen consisting of the insect image, text of whether the response was correct or incorrect, and the correct class was shown for 2 s followed by a 4–8 s fixation. The timing of the stimulus and feedback phases of the learning trials was jittered to optimize general linear modeling estimation of the fMRI data. Within one functional run, each of the eight insect images was presented in four learning trials. The order of the learning trials was pseudo randomized in blocks of 16 trials such that the eight stimuli were each presented twice. One functional run was 388 s in duration. Each of the learning problems included four functional runs for a total of 16 repetitions for each insect stimulus. The entire experiment lasted ~65 min

**Behavioral analysis**. Participant-specific learning curves were extracted for each problem by calculating the average accuracy across blocks of 16 learning trials. These learning curves were used for the computational learning model analysis. Furthermore, a mixed effect logistic regression analysis was performed on the

behavioral data. Specifically, fixed effects of stimulus repetition, problem complexity, and learning order along with random intercepts were estimated in predicting trial-by-trial accuracy across all participants. Accuracy improved across stimulus repetitions ($\chi^2 = 769.9$, $p < 0.0001$), differed between problem complexity overall ($\chi^2 = 970.1$, $p < 0.0001$), and changed differently across repetitions for the problems ($\chi^2 = 68.9$, $p < 0.0001$), but did not differ between learning orders ($\chi^2 = 2.087$, $p < 0.149$).

**Computational learning model.** Participant behavior was modeled with an established mathematical learning model, SUSTAIN[8]. SUSTAIN is a network-based learning model that classifies incoming stimuli by comparing them with memory-based knowledge representations of previously experienced stimuli. Sensory stimuli are encoded by SUSTAIN into perceptual representations based on the value of the stimulus features. The values of these features are biased according to attention weights operationalized as receptive fields on each feature attribute. During learning, these attention weight receptive fields, which change as a function of the latent model variable $\lambda_i$, are tuned to give more weight to diagnostic features. SUSTAIN represents knowledge as clusters of stimulus features and class associations that are built and tuned over the course of learning. New clusters are recruited, and existing clusters updated according to the current learning goals. A full mathematical formulization of SUSTAIN is provided in its introductory publication[8].

To characterize the attention weights participants formed during learning, we fit SUSTAIN to each participant's trial-by-trial learning behavior. First, SUSTAIN was initialized with no clusters and equivalent attention weights across the stimulus feature attributes. Then, stimuli were presented to SUSTAIN in the same order as a participant's experience, and model parameters were optimized to predict each participant's trial-by-trial responses in the three learning problems through a maximum likelihood optimization method[51]. Specifically, model likelihood was calculated based on the probability of the model making the same response as the participant in each trial and this likelihood was maximized through the differential evolution optimization algorithm provided in the *scipy* python library. In the optimization procedure, the model state at the end of the first learning problem was used as the initial state for the second learning problem. In doing so, parameters were optimized to account for learning with the assumption that attention weights, and knowledge clusters learned from the first problem carried over to influence learning in the second problem. Similarly, model state from the second problem carried over and influenced early learning in the third problem. Thus, problem order effects are considered a natural consequence of our model fitting approach. The optimized parameters were then used to extract measures of feature attribute attention weights throughout learning in the three problems. Specifically, for each participant, the model parameters were fixed to the optimized values and the model was presented with the trial order experienced by the participant. On each trial, the values of the feature attribute attention weights, $\lambda_i$, were extracted for each participant. This was repeated for each of the three learning problems. The average value and 95% confidence intervals of SUSTAIN's five free parameters were: $\gamma = 8.96 \pm 0.82$, $\beta = 1.51 \pm 0.34$, $\eta = 0.08 \pm 0.03$, $d = 17.04 \pm 2.05$, $\tau_h = 0.11 \pm 0.04$.

**MRI data acquisition.** Whole-brain imaging data were acquired on a 3.0T Siemens Skyra system at the University of Texas at Austin Imaging Research Center. A high-resolution T1-weighted MPRAGE structural volume (TR = 1.9 s, TE = 2.43 ms, flip angle = 9°, FOV = 256 mm, matrix = 256 × 256, voxel dimensions = 1 mm isotropic) was acquired for coregistration and parcellation. Two oblique coronal T2-weighted structural images were acquired perpendicular to the main axis of the hippocampus (TR = 13,150 ms, TE = 82 ms, matrix = 384 × 384, 0.4 × 0.4 mm in-plane resolution, 1.5 mm thru-plane resolution, 60 slices, no gap). High-resolution functional images were acquired using a T2*-weighted multiband accelerated EPI pulse sequence (TR = 2 s, TE = 31 ms, flip angle = 73°, FOV = 220 mm, matrix = 128 × 128, slice thickness = 1.7 mm, number of slices = 72, multiband factor = 3) allowing for whole-brain coverage with 1.7 mm isotropic voxels.

**MRI data preprocessing and statistical analysis.** MRI data were preprocessed and analyzed using FSL 5.0.9[52] and custom Python routines. Functional images were realigned to the first volume of the seventh functional run to correct for motion, spatially smoothed using a 3 mm full-width-half-maximum Gaussian kernel, high-pass filtered (128 s), and detrended to remove linear trends within each run. Functional images were registered to the MPRAGE structural volume using Advanced Normalization Tools, version 1.9[53].

**Neural compression analysis.** The goal of the neural compression analysis was to assess the informational complexity of the neural representations formed during the different learning problems. To index representational complexity, we measured the extent that neural activation patterns could be compressed into a smaller dimensional space according to PCA. The compression analyses were implemented using PyMVPA[54] and custom Python routines and were conducted on preprocessed and spatially smoothed functional data. First, whole-brain activation patterns for each repetition of each stimulus within each run were estimated using an event-specific univariate GLM approach[16]. This approach allowed us to model

estimates of neural patterns for the eight insect stimuli across the trials in each learning problem. For each classification problem run, a GLM with separate regressors for stimulus presentation on each trial, modeled as 3.5 s boxcar convolved with a canonical hemodynamic response function, was conducted to extract voxel-wise parameter estimates for each trial. In addition, two separate regressors for correct and incorrect feedback events (2 s boxcar) and two separate regressors for response events (impulse function at the time of response), as well as six motion parameters were included in the GLM. This modeling strategy targeted the neural representations specific to viewing the stimuli separate from processes associated with feedback events and trial outcomes for the participants' responses. This procedure resulted in, for each participant, whole-brain activation patterns for each trial in the three learning problems.

We assessed the representational complexity of the neural measures of stimulus representation during learning with a searchlight method[17]. Using a searchlight sphere with a radius of four voxels (voxels per sphere: 242 mean, 257 mode, 76 minimum, 257 maximum), we extracted a vector of activation values across all voxels within a searchlight sphere for all 32 trials within a problem run. These activation vectors were then submitted to PCA to assess the degree of correlation in voxel activation across the different trials. PCA was performed using the singular value decomposition method as implemented in the *decomposition. PCA* function of the *scikit-learn* (version 0.17.1) Python library. To characterize the amount of dimensional reduction possible in the neural representation, we calculated the number of PCs that were necessary to explain 90% of the variance ($k$) in the activation vectors. We scaled this number into a compression score that ranged from 0 to 1,

$$\text{compression} = 1 - \frac{k}{n}, \tag{1}$$

where $n$ is equal to 32, the total number of activation patterns submitted to PCA. By definition, 32 PCs will account for 100% of the variance, but no compression. With this definition of neural compression, larger compression scores indicated fewer PCs were needed to explain the variance across trials in the neural data (i.e., neural representations with lower dimensional complexity). In contrast, smaller compression scores indicated more PCs were required to explain the variance (i.e., neural representations with higher dimensional complexity). This neural compression searchlight was performed across the whole brain separately for each participant and each run of the three learning problems in native space. One limitation that is important to note is that this PCA approach to indexing neural compression does depend on the success of the single-trial GLM parameter estimates. Brain regions with lower signal or higher noise may lead to noisy single-trial parameter estimates that would inflate the PCA estimation of dimensionality (i.e., a higher number of PCs). For this reason, differences in neural compression are most interpretable when observed for within-subject factors evaluated within the same brain region, as has been done in the current work.

Group-level analyses were performed on the neural compression maps calculated with the searchlight procedure. Each participant's compression maps were normalized to MNI space using ANTs[53] and combined into a group dataset. To identify brain regions that demonstrated neural compression that was consistent with the representational complexity of the learning problems, we performed a voxel-wise linear mixed effects regression analysis using the *statsmodels* Python library (version 0.8). The mixed effects model included factors of problem complexity and learning block as fixed effects as well as participants as a random effect to predict neural compression. The interaction of problem complexity and learning block was the central effect of interest. We also included each participant's accuracy for the three problems within each learning block as a covariate. This regression model was evaluated at each voxel. A statistical map was constructed by saving the $t$-statistic of the interaction between complexity and learning block. The resulting statistical map was voxel-wise corrected at $p = 0.001$ and cluster corrected at $p = 0.05$, which corresponded to a cluster extent threshold of greater than 259 voxels. The cluster extent threshold was determined with AFNI[55] 3dClustStim (version 16.3.12) using the *acf* option, second-nearest neighbor clustering, and two-sided thresholding. The 3dClustSim software used was downloaded and compiled on November 21, 2016 and included fixes for the recently discovered errors of improperly accounting for edge effects in simulations of small regions and spatial autocorrelation in smoothness estimates[56]. Additional statistical maps of the main effects of problem complexity, learning block, and accuracy were also interrogated. No significant clusters were found for accuracy; see Supplementary Fig. 1 and Supplementary Table 1 for the results for problem complexity and learning block.

We assessed the nature of the interaction in the vmPFC cluster by extracting each participant's average neural compression score within the cluster for each problem across the four learning runs. This average compression is plotted in the middle panel of Fig. 2b. A linear mixed effects model estimated with Bayesian methods testing the same regression model as described above was performed on the neural compression scores from the peak voxel within the vmPFC cluster using the *rstanarm* (version 2.18.2) R library. Relative to the standard frequentist approach to linear regression, a Bayesian-estimated linear mixed effects approach estimates a full probability model that incorporates uncertainty estimates about the outcome and predictor variables within a hierarchical framework that explicitly models participant and group-level effects[57]. Through Monte Carlo Markov Chain (MCMC) procedures, a regression model can be estimated that provides credible

probability estimates for predictor variables without the constraints of normality that limit frequentist linear regression techniques. The regression models conducted here were based on default arguments for the *stan_glmer* function: weakly informed priors with regularization to prevent over-fitting and four MCMC chains of 2000 samples, the first half of which are discarded as "warm-up" samples. This results in 4000 total samples from the posterior distribution of the model. The posterior samples from each factor in the model can be used to assess model convergence, estimate the average factor coefficient, define a 95% high-density interval around each factor estimate (i.e., the Bayesian alternative of confidence intervals), and a P value representing the proportion of samples from each factor's posterior distribution that counter the sign of the mean estimate (i.e., this can be interpreted as a measure of significance similar to frequentist p values). Model convergence is assessed with the Rhat statistic which estimates the consistency between independent MCMC chains—values greater than 1.1 suggest the MCMC sampling did not converge. In all reported Bayesian-estimated linear mixed effects model here, the Rhat values were less than 1.1 suggesting model convergence.

It is important to note that separately analyzing neural compression from the peak voxel within the vmPFC cluster does not represent a set of independent findings. It does, however, provide a window into the nature of the factors underlying this cluster and the contribution (or lack thereof) of learning problem order and accuracy. Results from the Bayesian-estimated mixed effects model are summarized in Table 2. The posterior distributions for learning block, problem complexity, and their interaction are plotted in the right panel of Fig. 2b.

One potential issue with our findings is that our learning problems vary in their complexity, and task difficulty can change univariate neural activation. As such, the concern exists that rather than changes in neural compression, our findings are driven by simple changes in overall activation levels across problem complexity. To address this concern, we examined the mean activation in the vmPFC cluster across learning blocks and problem complexity with a Bayesian-estimated mixed effect linear regression. We found no differences in average activation across learning block ($\beta_{mean} = -20.3$, 95% HDI = $[-134.8, 89.2]$, $P = 0.72$), problem complexity ($\beta_{mean} = 25.1$, 95% HDI = $[-115.2, 163.8]$, $P = 0.73$), nor an interaction of these factors ($\beta_{mean} = -1.309$, 95% HDI = $[-82.7, 126.7]$, $P = 0.93$). Thus, in our paradigm, problem complexity did not lead to differences in overall neural activation that changed across learning.

One additional concern with our experimental procedure is that the task with the highest complexity was always learned first. This particular problem order was important for another purpose in a previous analysis of the data[28]. It is a valid concern that the differences in neural compression are driven simply by this first problem being the most difficult and least practiced. Such conditions could potentially be reflected in greater noise in the neural representations for this higher complexity task which might lead to lower neural compression, but for reasons not

due to problem complexity. To address this concern, we performed the whole-brain voxel-wise linear mixed effects regression analysis comparing complexity across learning blocks while controlling for problem order and accuracy, but only for the data associated with the low and medium complexity problems. By excluding the high complexity data, we can target the effect of complexity without the confound of learning order. If the interaction remains between problem complexity and learning block in predicting neural compression in the vmPFC region, it stands to reason that learning order is not a significant driver in the current findings. Indeed, this follow-up analysis revealed a very similar cluster in vmPFC (Fig. 5) and no other regions survived cluster correction.

We also performed a Bayesian-estimated linear mixed effects analysis on the peak voxel from the vmPFC cluster in Fig. 2b, but only for the data associated with the low and medium complexity problems. Again, we found similar results to the full dataset with a significant interaction between learning block and complexity ($\beta_{mean} = -0.021$, 95% HDI = $[-0.034, -0.008]$, $P = 0.0015$). Both of these analyses suggest that neural compression changes across learning blocks and increases more for the low vs. the medium complexity problem when restricted to a subset of the data with counterbalanced learning order.

**Category-specific coding in neural compression**. The data-driven nature of the PCA approach for neural compression can reveal the degree that neural patterns can be compressed, but not necessarily why this compression is possible. In the current study, the critical hypothesis is that learning to attend to problem-specific information will impact neural representations in a manner consistent with the representational complexity of the problem. Demonstrating that neural compression increases with learning and does so according to problem complexity provides compelling evidence in support of our hypothesis, but support that is nonetheless indirect.

To directly assess the contribution of category coding present in the compression findings, we analyzed how trials for a given problem within a learning block load onto the identified PCs. If category information is driving neural compression, trials with stimuli from the same category should load similarly on the PCs and trials from different categories should load differently on the PCs. In other words the distribution of trial loadings onto the PCs should discriminate between the two categories. We indexed the degree of category discrimination in PCA loadings by calculating the absolute difference between the average loadings within each category for the set of PCs identified in the neural compression analysis. These category loading differences were weighted by the explained variance of each PC and summed to create a measure of PC category discrimination (Fig. 4a). Higher values of category discrimination suggests that category coding is driving neural compression. On the other hand, if trials from both categories load similarly on the PCs, category discrimination would be equal to 0. A Bayesian-estimated mixed effects linear regression was conducted that evaluated the relationship between PC category discrimination and factors of learning block, problem complexity, learning order, and accuracy. We found that category discrimination was present in the PCA loadings (Fig. 4b and Table 3): not only did category discrimination increase with learning, it did so most for low followed by medium and high complexity problems.

**Relating neural compression with behavioral signatures of selective attention**. To evaluate the relationship between neural compression and model-based estimates of attention weighting, we first extracted individual participant-based measures of each. The participant-specific average neural compression within the vmPFC cluster was extracted for each learning problem. We used the SUSTAIN estimates of stimulus dimension attention weights, λ, to calculate a signature of selective attention. Throughout learning on trial-by-trial basis, SUSTAIN tunes attention weights based on the model parameters, the trial sequence, and the outcome of each trial. For each participant, we extracted the trial-by-trial derived attention weights in each learning problem based on the participant's best-fitting parameters. These attention weights for the three stimulus dimensions in each problem were transformed to sum to 1, thus creating a probability distribution representing the likelihood of attention to the three features. For example, given the

**Table 2 Results of the Bayesian-estimated linear mixed effects regression model predicting neural compression within the peak voxel of the vmPFC region depicted in Fig. 2b.**

|  | Estimate | 95% HDI | P |
|---|---|---|---|
| Intercept | 0.586 | 0.473, 0.699 | <0.001 |
| Block | 0.028 | 0.014, 0.041 | <0.001 |
| Complexity | 0.025 | 0.005, 0.044 | 0.017 |
| Accuracy | 0.046 | 0.001, 0.090 | 0.044 |
| Order | 0.069 | −0.076, 0.209 | 0.326 |
| Block:complexity | −0.013 | −0.019, −0.006 | <0.001 |

The mean estimated values, 95% high-density interval (HDI), and P values are reported for each fixed effect

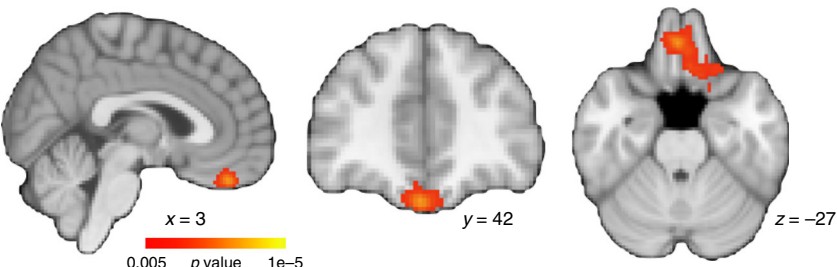

x = 3          y = 42          z = −27

0.005   *p value*   1e−5

**Fig. 5 A whole-brain voxel-wise linear mixed effects regression restricted to low and medium complexity problems (N = 23).** Similar to the main results in Fig. 2b, a cluster in vmPFC showed a significant interaction between learning block and problem complexity.

**Table 3 Results of the Bayesian-estimated linear mixed effects regression model of PC category discrimination across learning blocks and problem complexity.**

|  | Estimate | 95% HDI | P |
|---|---|---|---|
| Intercept | 0.0089 | −0.0056, 0.0234 | 0.111 |
| Block | 0.0080 | 0.0043, 0.0115 | <0.001 |
| Complexity | 0.0041 | −0.001, 0.0089 | 0.051 |
| Accuracy | −0.0029 | −0.0132, 0.0079 | 0.302 |
| Order | 0.0049 | −0.0002, 0.010 | 0.029 |
| Block:complexity | −0.0030 | −0.0047, −0.0013 | <0.001 |

The mean estimated values, 95% high-density interval (HDI), and P values are reported for each fixed effect

**Table 4 Results of the Bayesian-estimated linear mixed effects regression model predicting attention compression.**

|  | Estimate | 95% HDI | P |
|---|---|---|---|
| Intercept | −0.116 | −0.182, −0.049 | 0.001 |
| Block | 0.071 | 0.055, 0.087 | <0.001 |
| Complexity | 0.032 | 0.009, 0.054 | 0.006 |
| Accuracy | 0.103 | 0.055, 0.151 | <0.001 |
| Order | −0.025 | −0.054, 0.003 | 0.075 |
| Block:complexity | −0.028 | −0.035, −0.020 | <0.001 |

The mean estimated values, 95% high-density interval (HDI), and P values are reported for each fixed effect

**Table 5 Results of the Bayesian-estimated linear mixed effects regression model relating attention and vmPFC compression.**

|  | Estimate | 95% HDI | P |
|---|---|---|---|
| Intercept | −0.214 | −0.308, −0.126 | <0.001 |
| vmPFC comp. | 0.168 | 0.072, 0.277 | <0.001 |
| Block | 0.067 | 0.051, 0.083 | <0.001 |
| Complexity | 0.027 | 0.006, 0.049 | 0.017 |
| Accuracy | 0.093 | 0.045, 0.139 | <0.001 |
| Order | −0.031 | −0.064, 0.001 | 0.057 |
| Block:complexity | −0.026 | −0.033, −0.018 | <0.001 |

The mean estimated values, 95% high-density interval (HDI), and P values are reported for each fixed effect

compression was found when restricting the same regression analyses to only the low and medium complexity data ($\beta_{mean} = 0.210$, 95% HDI = [−0.082, 0.368], $P = 0.0005$).

**Reporting summary.** Further information on research design is available in the Nature Research Reporting Summary linked to this article.

## Data availability
The data collected for this study are available for download: https://osf.io/5byhb/. A reporting summary for this article is available as a Supplementary Information file.

## Code availability
Data analysis code available upon request.

attention weights [0.1, 0.1, 0.8], there is a probability of 0.8 that attention will be directed to the third stimulus dimension on any one trial. We then calculated entropy[39] across the attention weights for each problem separately:

$$\text{entropy} = -\sum_{i=1}^{3} a_i \log_2 a_i, \qquad (2)$$

such that $a_i$ is the attention weight for stimulus dimension $i$. This entropy measure indexes the dispersion of attention across the stimulus dimensions. If attention is unselective and all three stimulus dimensions are equally weighted, entropy is high. On the other hand, if attention is selective with the majority of weight on a single dimension, entropy is low. To better align attention entropy with our measure of neural compression, we transformed entropy into an index of attention compression:

$$\text{attention compression} = 1 - \text{entropy}/\log_2(1/3) \qquad (3)$$

Attention compression first scales entropy according to the maximum amount of entropy given three stimulus dimensions and then subtracts this ratio from 1. The result is a measure that ranges from 0 to 1 with low values corresponding to unselective attention and high values corresponding to more selective attention. As such, the attention compression index offers a unique signature for optimal attentional strategy across the three learning problems: the highest attention compression should be seen in the low complexity problem, an intermediate compression for the medium complexity problem, and the lowest attention compression for the high complexity problem. As a final step, for each participant, attention compression was averaged within learning block separately for each problem. The effect of problem complexity on attention compression was assessed with a Bayesian-estimated linear mixed effects regression that included factors of learning block, problem complexity, accuracy, learner order, and the interaction of block and complexity (see Fig. 3a and Table 4).

We next evaluated the relationship between vmPFC neural compression and attention weight entropy on an individual participant basis with Bayesian-estimated mixed effects linear regression. The regression model was conducted such that vmPFC neural compression, learning block, problem complexity, learning order, and accuracy were predictors of attention weight compression (Fig. 3b). This analysis estimates the degree that vmPFC compression is related to attention compression controlling for all of the other manipulated factors in the experiment, as well as individual differences in performance throughout learning. Finding that vmPFC compression is significantly related to attention compression, in spite of all of these other predictors, suggests that as attention evolves during learning (according to SUSTAIN's predictions of behavior), task-specific neural compression is evolving in the same fashion. Indeed, the results confirm this hypothesis showing a significant correspondence between vmPFC and attention compression (Table 5). The posterior distribution for the effect of vmPFC compression on attention compression reveals a robust finding (Fig. 3b, right panel). Importantly, a similar relationship between neural and attention

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

## Acknowledgements
Thanks to Christiane Ahlheim and Margaret Schlichting for paper comments. This work was supported by the Natural Sciences and Engineering Research Council (Discovery Grant RGPIN-2017-06753 to M.L.M.); National Institute of Mental Health (F32-MH100904 to M.L.M.; R01-MH100121 to A.R.P.); the Leverhulme Trust (RPG-2014-075 to B.C.L.); the Wellcome Trust (Senior Investigator Award WT106931MA to B.C.L.); and the National Institute of Child Health and Human Development (1P01HD080679 to B.C.L.).

## Author contributions
M.L.M., A.R.P. and B.C.L. designed the experiment and wrote the paper. M.L.M. conducted the research and data analysis.

## Competing interests
The authors declare no competing interests.
