## [Peer Review File · Nature Communications]

Reviewers' Comments:

Reviewer #1:

Remarks to the Author:

Mack and colleagues report interesting fMRI results from a human category learning task that lend support to the hypothesis that a medial PFC region (often labeled as vmPFC or OFC) is implicated in the compression of the external environment into goal-relevant state representations. The authors show that the compression of neural representations in this region (as measured by the numbers of principal components necessary to explain the variance in blockwise neural data) decreases over the course of learning different categorization rules. The extent of this decrease varies with the complexity of the learning problem, in that compression is highest for low complexity, as expected given that fewer features are required to correctly solve the low-complexity task. Overall this is a compelling set of results that adds to a growing picture of how the brain builds custom representations for solving tasks. I have a few major concerns, some requiring further analysis and some regarding presentation/interpretation of the data, and a few minor comments, as detailed below.

Major concerns:

First, my read of the methods suggests that the outcome for each trial (correct or incorrect) was not modeled in the GLM. This means that accuracy, or consistency of outcomes, can affect the activations for the different stimuli. Since more complex problems took longer to learn, per stimulus there would be more errors throughout the task (or at least in the first blocks). If outcome is an important variable encoded in mPFC, this may affect the observed compression (harder to compress representations that differ in outcomes). I would therefore like to see that the results hold when outcome is modeled in the GLM and therefore regressed out of the stimulus activations.

Second, I found the terminology of medial PFC too general — the mPFC includes wide areas of the PFC, all the way dorsally and bordering (and sometimes including) the ACC. It seems that the activations here are much more localized to the ventral mPFC or even medial OFC. I would welcome a more detailed discussion of what area was found, and also relating the hypothesis and findings to the literature more carefully (not to all literature on mPFC, but focusing on its ventral part).

Indeed, the results in this paper speak to an important gap in a growing literature that has implicated the OFC in the representation of states subserving reinforcement learning and decision-making. Specifically, they demonstrate that OFC not only represents the current state (as in Schuck et al., 2016), but also that this representation is dynamic, at least insofar as its complexity adapts to that of the learning problem. I do, however, think the discussion goes one step too far, beyond what the data show, when it claims that these findings "provide important evidence for a role of the mPFC during the formation of conceptual maps of experience" (line 248). We do not know from these results if mPFC forms/learns these representations, or rather it only reflects information learned elsewhere (e.g., in hippocampus, or dlPFC). It very well may be that mPFC learns the representations, but I do not think these data provide evidence of that. The discussion laments the fact that empirical work has failed to directly examine during cognitive map formation. However, the authors may have missed one paper that is closely related to work discussed here (even using similar analyses that measure the entropy of attention weights) and has done exactly that, which is Leong et al., "Dynamic Interaction between Reinforcement Learning and Attention in Multidimensional Environments", *Neuron* (2017).

Third, the authors use a well established model of attentional selection in category learning to make a link between the decrease in neural complexity and a model-based index of attentional breadth (the degree to which attention is distributed to over all features of the classification problem). Here, again, the discussion makes claims that go beyond what the data (or analyses) show. In particular, I am

referring to the claim that "the process of learning to compress in mPFC is consistent with the mechanisms of SUSTAIN" (line 216). Since there was no comparison with other models or mechanisms, we don't really know this. SUSTAIN is but one model that suggests more focused attention (and simpler representations) for simpler problems. I therefore find this claim unconvincing, and even somewhat misleading — no evidence for the specific assumptions of the SUSTAIN model were presented here.

Indeed, while the neural results are compelling, it would be interesting to more explicitly link them to patterns of behavior. Because model parameters were optimized for accuracy over 16-trial blocks (rather than choices), it is difficult to glean whether the learned attention weights reflect the actual learning process, or are tuned to coarsely match different levels of performance. Put differently, does the model actually tell us how participant learn these representations, or merely reflect the fact that they have?

Fourth, the order of the three types of problems is such that the most complex one always appeared first. This issue is acknowledged (line 554 onwards) and the authors took a remedy in analyzing mPFC only in the last two problems (1-dimension or 2-dimension), of which the orders were counter-balanced. One issue is that I believe this was using the same ROI they had identified with the searchlight based on interaction of learning block and problem complexity (with all the data). So there is a risk of double-dipping. It would be better to do the same whole-brain searchlight analysis with only these two problems and show that this still yields the mPFC alone.

It is stated that this experiment order was designed for a purposes described in the authors' previous paper. Skimming that paper, there the stated reason for having the most complex problem first was to use the first game as a familiarization task in order to "eliminate any neural activation due to stimulus and task novelty during the learning tasks". Indeed, there seemed to be no training task included in the instructions, so that purpose seems to be quite reasonable. However, this makes me worry about including here data that were designed to be discarded. Again, a full analysis of only the last two tasks (as suggested above) would solve this issue.

Finally, the result of increasing neural compression with task with fewer relevant features was taken to support the idea that mPFC filters irrelevant dimensions in learning. But a more direct line of evidence would be to show that the way variance in neural activity is compressed is consistent with the way the task can be compressed, i.e., that the patterns of trials that have the same feature in the relevant dimension should be more similar than those with different features on the relevant dimension, regardless of which was the irrelevant dimension. Citation 25, which analyzed the same data, seems to have used RSA but focused on hippocampus. It would be great to see a similar analysis of the mPFC/OFC area seen here.

Minor comments:

- Another reference that the authors may be interested in is Bar-Gad et al., "Reinforcement-driven dimensionality reduction--a model for information processing in the basal ganglia" (2000)

- I found the description of the modeling in the methods to be lacking. First, I understand that you don't want to describe the model in detail, but still — can you explain what are the parameters of the model and give some intuition for them? Understanding the degrees of freedom that the model has to explain behavior will go a long way to unpacking it. Second, what is a maximum likelihood genetic algorithm optimization method (line 422) and how was it implemented? How was the likelihood calculated, given that you are not assigning a probability to each decision (or are you?). If I wanted to replicate the analyses here, I don't think the level of detail provided is sufficient.

- Continuing on the above issue: Lines 401-437 discuss the attention weights calculated from the SUSTAIN model, which are the basis of attentional entropy in Fig 3C. From the description it is not clear how the attention weight is calculated from the parameters whose values are reported in lines 436-437. Going beyond what a reader of the paper would do, I read the SUSTAIN paper to the part of the definition of those parameters. It seems that none of these parameters can directly be taken as attentional weights. Therefore, it may be helpful to explain how attentional weight is derived from those parameters.

- It was unclear from the main text what the timescale of the attention weights was, relative to that of the neural complexity measure. If I understood the modeling procedure correctly, these are weights obtained by fitting the accuracy data from all learning problems jointly, and then using the parameters to extract (average?) weights from the last quarter of each categorization problem. First — are attention weights averaged over that last quarter, or taken at the timepoint of the beginning of that quarter (as the text suggests)? Second, does this match the time bin used for averaging of the complexity measure depicted in Figure 3?

- From my understanding of the SUSTAIN model, the clustering it learns seems to imply an assumption that the environment is static - the classes should not change over time. But lines 423-429 mention that what the model learns from one problem carries over to the next problem (which has a different classification rule, and also a different set of labels, e.g., from warm-cold to maybe eastern-western). It is not intuitive how exactly the learned cluster can be carried to the next problem in the SUSTAIN model.

- In the methods on page 13 (line 401), should the title of the sub-section be "computational modeling of learning"?

- Line 545: "changes in" appeared twice.

- Line 307-310. The first sentence mentioned legs was one dimension but not tails. But the second sentence mentioned pointy or rounded tail without mentioning legs as one dimension. They seem to conflict.

- Line 250: important -> importance

Reviewer #2:

Remarks to the Author:

This is a super cool paper that tests the proposal that neural representations of external events – evident in the pattern of BOLD response – adjust to reflect the "goal-relevant" features of those events. The authors measured BOLD signal in people learning to classify insects that differed along 3 dimensions. In one trial block or task, only one dimension was important. In another, two dimensions were critical. In a third, all three were relevant. The authors then asked how many dimensions in a PCA analysis were required to account for 90% of the variance in BOLD response in different brain regions, across learning in the 3 conditions. They found that only one area – mPFC – showed an interaction between the number of PCA dimensions required in the three blocks and the learning in the blocks. This result is novel I think because it shows that the patterns in human PFC are more complex based on the demands of the task. While this has been suggested before, this report is one of only a couple I can think of that show this in a procedure in which the cues are the same, the basic task performance is the same, and there is no order of training effect. This is very nice. Additionally this is

the only study I know of that shows this is an effect that occurs during learning in the PFC. That is the compression of information that happens in the simpler variants occurs as the information becomes relevant. This is one of several very important open questions of this general idea I think. This study addresses it very nicely. I have many questions but basically I think the experiment is beautiful and the results are clear and important and should be published.

Ok so on to my questions. Probably my most important one is to ask the authors to provide more detail on a number of things.

First I don't know what attention weight entropy is, though I can imagine a bit, nor do I recognize SUSTAIN. These things should be explained so I can understand them in the context of the experiment.

Second I appreciate the significance of the interaction identified. And that it is only in mPFC. Though I have to say I always find it sort of amazing when fMRI studies find something in just one area. I think it might mitigate some skepticism here if the authors could describe what is in other areas. And the threshold for this claim. For example, where there are other areas that showed learning effects but no difference for complexity or vice versa? I (and the authors) would be particularly interested in what hippocampus is doing. Is it not changing at all?

Third have the authors considered expressing their results in any other way than dimensionality reduction? I would think the basic approach they are using would also predict dramatic differences in classification accuracy with the MVPA patterns? That is, if you used the patterns early on to classify the insects into their 9 categories, you would get good classification. This classification would not change for the high dimension blocks, whereas it would degrade and in interesting ways in the lower dimension blocks? Right? Showing this in more than this one simple way would dramatically increase the impact of the results I think. I'd also be interested in how misclassification tracks with errors.....

Fourth, the authors say there is no effect of order. This is nice for their conclusion, but I wonder why this is. It seems to me that I might have expected some effect of prior block on where the initial compression value is on a new block. Instead this implies that the system resets or something. Is this an artifact of how the blocks were separated? If not, I am wondering if there is more information here. In particular, I think it would be interesting to show that the compression moves in both directions. That is, not only does it go down, if task relevance does not require information. But that once it has gone down, it takes a bit of time to go back up. Likewise where does it start a priori? In the very first block, is there a reduction initially even when I am presented with the high dimension task first? It seems like there should be an initial reduction because there are many bits of information available that I must learn to ignore in any task..... The answers to these questions do not impact the headline finding, but I think they are questions that are important and the authors are in a position to answer them.

Finally I think I have a small objection to something in the discussion. The authors say the data support the proposal that "latent mPFC representations are goal-specific". I am not sure what the authors mean by latent and how they have shown this. To my mind, a latent representation would actually be one for the task irrelevant features in the low dimensionality task. Showing it would require demonstrating some more rapid appearance of its influence in a high dimensionality block..... I think the authors have the data to address this and it would be extremely interesting to know if this occurs. But I think they are not showing this. If they could either clarify what they mean or remove this term, I'd appreciate it.

Here are two papers that came to mind when reading that are not cited. Possibly they are not relevant

to the authors, but just in case:

Guise, K. G. and M. L. Shapiro (2017). "Medial prefrontal cortex reduces memory interference by modifying hippocampal encoding." *Neuron* 94: 183-192.

Wikenheiser, A. M., et al. (2017). "Suppression of ventral hippocampal output impairs integrated orbitofrontal encoding of task structure." *Neuron* 95: 1197-1207.

Reviewer #3:

Remarks to the Author:

The present work uses a modeling approach to estimate the dimensionality of data represented in cortical regions and how it changes with learning, with the goal of finding cortical regions that compress the nature of the representation. A clever design was used that requires decisions based on 1,2, or 3 dimensions of a stimulus so as to set up a framework for detecting change in dimensionality.

That said, a basic challenge of the manuscript and findings is that the modeled results yield patterns of response that have many assumptions built in and are derived variables quite far from the data. On the one hand, I appreciate that is the power of model approaches to the degree the model captures processes and components that are not available in simpler manifestations of the data. On the other hand, modeled data brings with it a burden to unpack and help the reader digest what they are seeing.

Even with several passes through the manuscript I had great difficulty following what the measures meant. I also had concerns that there are forms of over fitting and double dipping that have been prevalent in linear models. Some examples are listed below to help the authors see the data from perspective of an outside first-time reader.

(1) If I understand the model correctly, it is based on the complexity of the BOLD response to the model that considers the multiple trials, with maximal complexity (dimensionality) being the case where there are responses to each of the trials that are different from one another. This would require a large set of dimensions to account for the variance in the BOLD signal patterns. Regions that could have 'compression' are necessarily those with minimal variable responses (minimal paradigm-locked response or constant response could both yield low dimensionality).

Here is where I start to lose track of the relation between the model and the data. The sought property is an interaction of the dimensionality (as operationalized by the number of principal components that capture 90% of the variance) between the learning condition and learning block. This is a challenging model and unit to wrap ones head around, not simply because of its complexity but because of some ambiguities. For example, do high noise regions always have high PCA numbers because the 90% threshold is not achieved? Do regions with low response also always bend towards high dimensionality? Given change in dimensionality is the quantity being captured in compression, what is the actual change in signal? Does a robust highly variable signal converge to a more constant but clear response? Does the response modulation just go away? Is it diverse then becomes stationary across trials? Put simply, the operational definition of 'compression' on p. 5 leaves a great deal of the nuances of the data beneath the surface.

(2) In a few places I worried that there was room for fishing and circularity in the estimated parameters and their certainty. In the methods, on p. 16, it is appropriately noted that the clusters were defined by the model and that post hoc analysis does not represent independent findings.

However, the display of certainty and confidence intervals throughout, much like displaying biased r correlation values or inflated p values in traditional circular analyses, may be problematic. Take the '95% confidence bands' in the shaded plot within Figure 2B. Isn't that confidence interval derived from the idiosyncratic region defined in this sample that has been selected from a large number of possible regions? As such, the confidence bands, much like circularly defined p values, are almost certainly not an unbiased population estimate. Is this a problem for all of the data and scatter and confidence bands reported throughout the paper? Is the p value (10^{-5}) reported on p. 6 for the interaction derived from a biased selection of the region of 653 voxels chosen to have the tested property? I am unsure if there is or if there is not an issue in all cases, but the circular procedures (define map with a large number of mapped voxels, then select region representing a subset of voxels, then quantify behavior in that selected subset) smack of the kind of approach that has gotten the field into trouble in the past.

(3) Given all of the assumptions of the model and the correlational nature of the data, the claims are too strong. The strong claim in the title that MPFC is active in compressing the concept representation is one such example.

Reviewer 1

Reviewer 1 summarised our study as: “Overall this is a compelling set of results that adds to a growing picture of how the brain builds custom representations for solving tasks.” We thank the Reviewer for this encouraging view of our work. Additionally, Reviewer 1 had several criticisms and suggestions for improving our analysis, description of methods, and interpretation. By addressing these thoughtful comments, our revision offers an empirically stronger demonstration of learning-related neural compression.

R1.1: *“First, my read of the methods suggests that the outcome for each trial (correct or incorrect) was not modeled in the GLM. This means that accuracy, or consistency of outcomes, can affect the activations for the different stimuli. Since more complex problems took longer to learn, per stimulus there would be more errors throughout the task (or at least in the first blocks). If outcome is an important variable encoded in mPFC, this may affect the observed compression (harder to compress representations that differ in outcomes). I would therefore like to see that the results hold when outcome is modeled in the GLM and therefore regressed out of the stimulus activations.”*

Response: The Reviewer makes an excellent point, especially given that ventromedial prefrontal cortex (vmPFC) has been linked to reward and outcome processing. Because of this, we estimated trial-specific beta volumes based on the stimulus onset and included additional regressors for feedback events (separately for correct and incorrect feedback). In doing so, we are able to regress out activation related to different trial outcomes and target the neural representations elicited by the onset of the stimuli for the compression analyses. We have clarified the methods of our trial-specific beta estimation on pg. 17.

R1.2: *“Second, I found the terminology of medial PFC too general — the mPFC includes wide areas of the PFC, all the way dorsally and bordering (and sometimes including) the ACC. It seems that the activations here are much more localized to the ventral mPFC or even medial OFC. I would welcome a more detailed discussion of what area was found, and also relating the hypothesis and findings to the literature more carefully (not to all literature on mPFC, but focusing on its ventral part).”*

Indeed, the results in this paper speak to an important gap in a growing literature that has implicated the OFC in the representation of states subserving reinforcement learning and decision-making. Specifically, they demonstrate that OFC not only represents the current state (as in Schuck et al., 2016), but also that this representation is dynamic, at least insofar as its complexity adapts to that of the learning problem. I do, however, think the discussion goes one step too far, beyond what the data show, when it claims that these findings “provide important evidence for a role of the mPFC during the formation of conceptual maps of experience” (line 248). We do not know from these results if mPFC forms/learns these representations, or rather it only reflects information learned elsewhere (e.g., in hippocampus, or dlPFC). It very well may be that mPFC learns the representations, but I do not think these data provide evidence of that. The discussion laments the fact that empirical work has failed to directly examine during cognitive map formation. However, the authors may have missed one paper that is closely related to work discussed here (even using similar analyses that measure the entropy of attention weights) and has done exactly that, which is Leong et al., “Dynamic Interaction between Reinforcement Learning and Attention in Multidimensional Environments”, Neuron (2017).”

Response: We agree with the Reviewer that our finding is much more localized within mPFC. We have changed references to this region throughout the revised manuscript to reflect the vmPFC subregion and focus our discussion of related work to findings from vmPFC.

We also agree with the Reviewer's suggestion that the neural compression findings, as reported in the initial submission, only indirectly point to a role for vmPFC in building conceptual maps. In response to this limitation, we conducted a novel analysis that directly characterizes the structure of neural compression in vmPFC by indexing the degree of category-specific discrimination present in the PCA results. Specifically, we evaluated how trials for a given task and learning block loaded on the principal components (PCs) identified in the compression analysis. The logic of this analysis is that if vmPFC representations are tuned by attention according to task demands such that diagnostic dimensions are weighted more and nondiagnostic dimensions are weighted less, trials with stimuli from the same category should load similarly on the PCs and these loadings should differ between categories. For example, if all category A trials load positively on a PC and all category B trials load negatively on the same PC, this PC will transform the neural data to increase the discriminability of two categories. If, on the other hand, the compression provided by PCA is based on shared variance irrelevant to the learning task, trials will load on the PCs in a similar fashion regardless of their category. If categories are represented in the structure of the PCA loadings as a consequence of learning, category discriminability should increase over learning blocks and vary with task complexity. Indeed, this is exactly what we find, as can be appreciated in Figure 4 in the revised manuscript. As participants learned, trials from the same category were more likely to load onto the same principal components and these loadings differed between categories as a function of the problem complexity. We believe this new analysis speaks to concerns raised by the Reviewer (as well as the other two reviewers) about the nature of the neural compression findings and provides more direct evidence of problem-specific dimensionality reduction in vmPFC representations.

Additionally, we have significantly extended our analysis linking vmPFC compression to the attention weight predictions of SUSTAIN. More details are provided below (see response to R1.3). Briefly, we examined the correspondence between SUSTAIN and neural compression at a finer timescale and find that throughout learning 1) higher neural compression is associated with higher attention compression and 2) higher attention compression is related to greater category discrimination in the PCA loadings.

We see all these findings as evidence of a tight coupling between a theoretically-driven behavioural and neural index of dimensionality reduction due to new learning. And, that vmPFC is the only region that shows these effects throughout learning suggests it plays an important role in building structured knowledge. It is true that the Reviewer's suggestion that vmPFC may be reading out representations learned elsewhere cannot be fully ruled out and we have updated our discussion to reflect this limitation. More generally, we have tempered our claims throughout, including a slight change to the title, and offer the role of vmPFC in learning as an important hypothesis consistent with our findings that should be tested in future studies.

We also thank the Reviewer for suggesting the Leong et al. paper that we missed. We now include this relevant study in our discussion (pg. 10). In this paper, the authors looked at attention's influence on learning in a task that required participants to learn which one out of three stimulus dimensions was associated with higher reward probability. Across repetitions of this task, they found that the expected value of participant choices as determined by a reinforcement learning (RL) model that incorporates empirically-derived trial-by-trial attention weighting in both choice and update computations better accounted for behaviour and

corresponded with univariate activation in vmPFC. This finding suggests that attention is a critical actor during learning. The current findings build on this work in two important ways: 1) Whereas the Leong et al. findings are based on trial-by-trial univariate activation in vmPFC, we focus on the goal-directed nature of neural representations and how such goal-specific coding arises across learning. 2) We characterize how different levels of informational complexity are learned and reflected in the structure of neural representation. The informational complexity of the learning task in the Leong et al. study was constant and similar to the low complexity problem in our study in that one out of three dimensions was more informative. In the current work, we characterize how neural representations reflect learning in multiple tasks that require attending to and integrating across 1, 2, or 3 stimulus dimensions. We think these findings both extend prior work on reinforcement learning into the domain of concept learning and motivate a new mechanistic explanation for the role of vmPFC in guiding learning to goal-relevant features at the level of representation.

R1.3: *“Third, the authors use a well established model of attentional selection in category learning to make a link between the decrease in neural complexity and a model-based index of attentional breadth (the degree to which attention is distributed to over all features of the classification problem). Here, again, the discussion makes claims that go beyond what the data (or analyses) show. In particular, I am referring to the claim that “the process of learning to compress in mPFC is consistent with the mechanisms of SUSTAIN” (line 216). Since there was no comparison with other models or mechanisms, we don’t really know this. SUSTAIN is but one model that suggests more focused attention (and simpler representations) for simpler problems. I therefore find this claim unconvincing, and even somewhat misleading — no evidence for the specific assumptions of the SUSTAIN model were presented here.*

Indeed, while the neural results are compelling, it would be interesting to more explicitly link them to patterns of behavior. Because model parameters were optimized for accuracy over 16-trial blocks (rather than choices), it is difficult to glean whether the learned attention weights reflect the actual learning process, or are tuned to coarsely match different levels of performance. Put differently, does the model actually tell us how participant learn these representations, or merely reflect the fact that they have?”

Response: We thank the Reviewer for pushing us on this point. As the Reviewer suggests, a potentially more informative approach is to fit the model such that it accounts for specific trial-by-trial behavioural responses rather than summary measures. In fact, this is the standard approach for current and new projects in our respective labs and an approach that SUSTAIN is well-suited for. In the revised manuscript, we have replaced the model analysis with a full reanalysis that follows a trial-by-trial approach. To be clear, the modeling analysis of behaviour was changed to be trial-by-trial, but the neural compression analyses remain at the level of learning block. Thus, the model is fit to predict choices on each trial and the resulting model measures are then summarized and related to neural compression for each learning block. The new modeling approach demonstrates three critical findings that extend beyond our initial submission:

- 1) Rather than limiting to only the final learning block as in the initial submission, we now examine SUSTAIN-based predictions of attention weighting throughout learning. To do so, we compare the average trial-by-trial attention weights within a learning block across problems and learning blocks (we average within learning blocks to temporally align with the neural compression measure). We find that SUSTAIN’s predictions for how attention

weights change throughout learning are consistent with changes in neural compression (see Figure 3A).

- 2) The revision also includes a new regression analysis that evaluates how attention weight changes are related to vmPFC compression. This analysis shows that even after accounting for learning block, problem complexity, problem- and block-specific behavioural accuracy, and the order of learning the problems, SUSTAIN-based attention weighting is significantly correlated with neural compression in vmPFC (see Figure 3B).
- 3) As mentioned in our response to R1.2, we performed a new analysis to characterize the structure of neural compression in vmPFC finding that PCA loadings discriminate categories across learning according to problem complexity (Figure 4). We further assessed the link between this compression-based measure of category discriminability and SUSTAIN-based attention weights throughout learning in a regression analysis. We find that higher attention compression is associated with higher category discriminability after accounting for learning block, problem complexity, learning accuracy, and problem order. This relationship closely links the mechanisms of attention biasing in SUSTAIN to goal-relevant neural compression and supports the claim for dimensionality reduction in vmPFC.

We believe that these three new findings presented in the revision serve as strong evidence that neural compression in vmPFC unfolds in a similar manner to SUSTAIN's learning mechanisms, in that goal-relevant information is given representational priority over irrelevant features.

It is true that we do not compare SUSTAIN to other models, as this is not the central focus of the study. However, SUSTAIN accounts for a wide range of learning behaviour and has been linked to specific neural mechanisms during learning (e.g., Mack et al., 2016; Davis et al., 2012). SUSTAIN is a quasi-neural network model that learns and predicts categorization behaviour based on a dynamic interaction of several theoretically-motivated mechanisms, one of which is attentional selection. Specifically, SUSTAIN learns which stimulus dimensions are most informative to given problem through a trial-by-trial delta learning rule similar to those used in RL models. However, SUSTAIN extends beyond typical RL models in that it explicitly represents combinations of attention-weighted features in an intermediate layer. These feature combinations (i.e., clusters in SUSTAIN terminology) are created and updated as a function of prediction error and, thus, adapt to the current task goals. For example, in accounting for the learning problems in the current study, SUSTAIN's modal solutions include 2, 4, and 8 clusters for the low, medium, and high complexity problems, respectively. In other words, SUSTAIN capitalizes on mechanisms of selective attention and clustering to perform representation learning such that multidimensional structures are efficiently learned; such learning remains a challenge for RL frameworks (Radulescou, Niv, & Ballard, 2019). As such, SUSTAIN serves as a credible theoretical framework for linking behavioural and neural measures of concept learning.

In our analysis, there is an implicit comparison in that SUSTAIN's predictions simply could have *not* related to neural compression. Moreover, in both the initial submission and revised model analyses, learning block accuracy for each participant in each problem was included as a covariate, providing a simple "model" of learning behaviour. Thus, SUSTAIN provides a window into learning that extends beyond simply tracking behavioural performance. In our view, that participant-specific quantitative predictions from SUSTAIN correspond with individual differences in neural compression during learning offers compelling and novel evidence that the

machinery of SUSTAIN is a worthwhile candidate for describing human learning and the type of dimensionality reduction that may underlie our vmPFC findings.

In the revised discussion (pg. 11), we now explicitly note the limitations to comparing neural measures with only a single model and suggest that neural compression may offer a novel index for model comparison in future studies.

R1.4: *“Fourth, the order of the three types of problems is such that the most complex one always appeared first. This issue is acknowledged (line 554 onwards) and the authors took a remedy in analyzing mPFC only in the last two problems (1-dimension or 2-dimension), of which the orders were counter-balanced. One issue is that I believe this was using the same ROI they had identified with the searchlight based on interaction of learning block and problem complexity (with all the data). So there is a risk of double-dipping. It would be better to do the same whole-brain searchlight analysis with only these two problems and show that this still yields the mPFC alone.”*

Response: The Reviewer has a fair point and an excellent suggestion for checking how well our findings generalize without concerns of problem order. Following the Reviewer’s suggestion, we have conducted the analysis with the first task excluded leaving only the low and medium complexity tasks, the order of which were counterbalanced. This analysis shows that restricting to only these problems reveals a very similar vmPFC region (see Fig. 5). We have also rerun the analyses linking neural compression to attention weights changes with the first problem excluded and find equivalent results (pg. 7). We thank the Reviewer for suggesting this important control analysis and think that the results strengthen our claims.

R1.5: *“Finally, the result of increasing neural compression with task with fewer relevant features was taken to support the idea that mPFC filters irrelevant dimensions in learning. But a more direct line of evidence would be to show that the way variance in neural activity is compressed is consistent with the way the task can be compressed, i.e., that the patterns of trials that have the same feature in the relevant dimension should be more similar than those with different features on the relevant dimension, regardless of which was the irrelevant dimension. Citation 25, which analyzed the same data, seems to have used RSA but focused on hippocampus. It would be great to see a similar analysis of the mPFC/OFC area seen here.”*

Response: The Reviewer makes a great suggestion. As noted above, our revision includes a new analysis that directly targets how neural patterns across trials are compressed in our PCA approach. This analysis demonstrates that trials within the same category load similarly onto the principal components that explain the most variance in the neural patterns and that these loadings differ between categories. This measure of category discrimination in neural compression increases with learning and depends on problem complexity (see Figure 4). Furthermore, the degree of category discrimination is related to predictions of attention weighting from SUSTAIN. These results suggest that the compression we see in vmPFC is tied to the dimensional structure of the learning problems and provides evidence that irrelevant features are being filtered at the level of representation.

It is worth noting that as part of the review process for our 2016 paper that the Reviewer mentions, we performed a whole-brain RSA searchlight to uncover neural representations that matched SUSTAIN-based predictions of task-relevant stimulus similarity. Although some regions including angular gyrus, insula, ventral temporal cortex, and striatum showed

consistency with SUSTAIN representations, none of these regions survived whole-brain cluster correction. It was these “non-findings” that motivated the current study—rather than learning representations that carefully integrate physical similarity and task-relevant attention weighting, as we saw in the hippocampus, we hypothesized that other regions may be coding more abstract representations divorced from physical similarity. This line of thought led to the current data-driven approach of neural compression.

R1: Minor comments

- *Another reference that the authors may be interested in is Bar-Gad et al., "Reinforcement-driven dimensionality reduction--a model for information processing in the basal ganglia" (2000)*

Response: Thank you for this reference, we have added it to the revision.

- *I found the description of the modeling in the methods to be lacking. First, I understand that you don't want to describe the model in detail, but still — can you explain what are the parameters of the model and give some intuition for them? Understanding the degrees of freedom that the model has to explain behavior will go a long way to unpacking it. Second, what is a maximum likelihood genetic algorithm optimization method (line 422) and how was it implemented? How was the likelihood calculated, given that you are not assigning a probability to each decision (or are you?). If I wanted to replicate the analyses here, I don't think the level of detail provided is sufficient.*

Response: We have expanded our description of the modeling methods (pg. 15).

- *Continuing on the above issue: Lines 401-437 discuss the attention weights calculated from the SUSTAIN model, which are the basis of attentional entropy in Fig 3C. From the description it is not clear how the attention weight is calculated from the parameters whose values are reported in lines 436-437. Going beyond what a reader of the paper would do, I read the SUSTAIN paper to the part of the definition of those parameters. It seems that none of these parameters can directly be taken as attentional weights. Therefore, it may be helpful to explain how attentional weight is derived from those parameters.*

Response: We thank the Reviewer for nothing this critical missing section of our methods. The attention weights are actually a product of the learning process and depend on a combination of the model parameters and trial sequence. Our revision includes an explanation of how the attention weights were obtained from SUSTAIN (pgs. 15 and 22).

- *It was unclear from the main text what the timescale of the attention weights was, relative to that of the neural complexity measure. If I understood the modeling procedure correctly, these are weights obtained by fitting the accuracy data from all learning problems jointly, and then using the parameters to extract (average?) weights from the last quarter of each categorization problem. First — are attention weights averaged over that last quarter, or taken at the timepoint of the beginning of that quarter (as the text suggests)? Second, does this match the time bin used for averaging of the complexity measure depicted in Figure 3?*

Response: As mentioned above in R1.3, in the revised analyses, we investigate attention weight predictions throughout learning. For calculating the attention compression measure, we

take the average attention weights within each block separately for each problem. As the Reviewer alludes to, this average of attention weighting for a learning block best aligns temporally with the GLM estimation for beta volumes and neural compression measures within each block. We have clarified this point in the Methods (pg. 15).

- From my understanding of the SUSTAIN model, the clustering it learns seems to imply an assumption that the environment is static - the classes should not change over time. But lines 423-429 mention that what the model learns from one problem carries over to the next problem (which has a different classification rule, and also a different set of labels, e.g., from warm-cold to maybe eastern-western). It is not intuitive how exactly the learned cluster can be carried to the next problem in the SUSTAIN model.

Response: To best capture the participants' learning experience, we did not re-initialize the model from one problem to the next. We assumed that what the participants learned from one task, specifically which dimensions matter more, would influence learning on the subsequent task, at least initially. In terms of how this plays out in the model, the carryover from task to task requires the model to unlearn attentional biases and adjust or replace clusters that initially match with stimulus features but not on the label. This mismatch leads to large updates in the clusters and attention weights and the carryover from the prior task is quickly unlearned. In our initial behavioural piloting for this study, we found that including carryover (which really just consists of not reinitialize the model between problems) significantly improved model fits. We have taken this strategy since.

- In the methods on page 13 (line 401), should the title of the sub-section be "computational modeling of learning"?

Response: We thank the Reviewer for noting this mistake. It is corrected in the revision.

- Line 545: "changes in" appeared twice.

- Line 307-310. The first sentence mentioned legs was one dimension but not tails. But the second sentence mentioned pointy or rounded tail without mentioning legs as one dimension. They seem to conflict.

- Line 250: important -> importance

Response: We thank the Reviewer for a careful read of our manuscript. These errors have all been corrected.

Reviewer 2

Reviewer 2 was very encouraging noting that the initial manuscript was "a super cool paper", that the "experiment is beautiful", and that "the results are clear and important and should be published". The Reviewer's kind words are certainly appreciated. In answering Reviewer 2's questions, we have significantly extended the modeling description and methods; added additional brain maps and tables showing compression effects related to the main effects of learning, complexity, and learning accuracy; and tempered our claims to best align with the

results we present.

R2.1: *“First I don’t know what attention weight entropy is, though I can imagine a bit, nor do I recognize SUSTAIN. These things should be explained so I can understand them in the context of the experiment.”*

Response: We have significantly expanded our description of the computational model and the modelling analyses we performed (pgs. 15 and 22). We have also converted attention weight entropy (which scaled such that larger values meant more equal weighting across feature dimensions) to a measure we call attention compression. Attention compression is a simple linear transform of entropy but represents attention weight differences such that larger values correspond with greater selectivity of diagnostic dimensions. This change in nomenclature and inversion of the attention measure is more intuitive and directly maps to our measure of neural compression.

R2.2: *“Second I appreciate the significance of the interaction identified. And that it is only in mPFC. Though I have to say I always find it sort of amazing when fMRI studies find something in just one area. I think it might mitigate some skepticism here if the authors could describe what is in other areas. And the threshold for this claim. For example, where there other areas that showed learning effects but no difference for complexity or vice versa? I (and the authors) would be particularly interested in what hippocampus is doing. Is it not changing at all?”*

Response: We agree with the Reviewer that full transparency here is key. To alleviate the concerns the Reviewer alludes to, we have included in the Supplement brain maps and tables for regions showing effects of learning block and complexity (Supplementary Figure 1 and Supplementary Table 1). We also note these effects in the main text (pg. 6). In our prior paper (Mack, Love, Preston, 2016), we demonstrated that the hippocampus stores representations consistent with task-specific attention-weighted stimulus similarity as predicted by SUSTAIN.

R2.3: *“Third have the authors considered expressing their results in any other way than dimensionality reduction? I would think the basic approach they are using would also predict dramatic differences in classification accuracy with the MVPA patterns? That is, if you used the patterns early on to classify the insects into their 9 categories, you would get good classification. This classification would not change for the high dimension blocks, whereas it would degrade and in interesting ways in the lower dimension blocks? Right? Showing this in more than this one simple way would dramatically increase the impact of the results I think. I’d also be interested in how misclassification tracks with errors.....”*

Response: The Reviewer raises an important point and one that the other Reviewers also noted. Namely, although the compression findings are consistent with behavioural and model-based changes in learning, they do not necessarily arise from representations of task structure. Our revision directly addresses this point with a new analysis that evaluates the degree of category discrimination present in the compression results (see Figure 4). Specifically, the trials for a given problem load on the principal components in a manner that discriminates between categories. This effect emerges over learning and depends on the complexity of the problem. Thus, even though PCA is entirely data driven, this new analysis demonstrates that the category structure underlying the different problems is driving neural compression. Collectively, the neural compression findings, their link to behaviour through the computational model predictions

of attention weighting, and the category discriminability evident in compression strongly support the claim for learning-mediated dimensionality reduction.

R2.4: *“Fourth, the authors say there is no effect of order. This is nice for their conclusion, but I wonder why this is. It seems to me that I might have expected some effect of prior block on where the initial compression value is on a new block. Instead this implies that the system resets or something. Is this an artifact of how the blocks were separated? if not, I am wondering if there is more information here. In particular, I think it would be interesting to show that the compression moves in both directions. That is, not only does it go down, if task relevance does not require information. But that once it has gone down, it takes a bit of time to go back up. Likewise where does it start a priori? In the very first block, is there a reduction initially even when I am presented with the high dimension task first? It seems like there should be an initial reduction because there are many bits of information available that I must learn to ignore in any task..... The answers to these questions do not impact the headline finding, but I think they are questions that are important and the authors are in a position to answer them.”*

Response: The Reviewer’s question reveals an interesting aspect of our results. When we explicitly included order as a factor in our neural compression analysis, we found no reliable effect of order. Likewise, analyses of attention compression across learning blocks also show no effect of order. However, during behavioural piloting of this task, we found that modelling the problems in a continuous fashion such that model representations and attention weights from the last problem were carried over to the next problem improved the overall model fits relative to an alternative account wherein the model was completely reinitialized between problems. Given this initial pilot result, we have used the across-problem carry over approach in analyzing the current data. Thus, although we find no differences due to problem order at the time scale of learning blocks, we do see small effects of order at the level of trial-level model fits. This difference is likely due to the differences in time scale; when switching to a new problem, participants (and model) are influenced by the last problem’s compression and attention weights but these are quickly unlearned and updated within the first few trials of the new problem. Thus, what the Reviewer suggest may be happening, wherein higher compression follows lower compression and vice-versa, is actually happening in our experiment when switching between the low and medium complexity problems. However, we don’t have the power in the current study to examine trial-level effects of problem carryover. This is a very interesting idea and one we hope to pursue in future work.

R2.5: *“Finally I think I have a small objection to something in the discussion. The authors say the data support the proposal that “latent mPFC representations are goal-specific”. I am not sure what the authors mean by latent and how they have shown this. To my mind, a latent representation would actually be one for the task irrelevant features in the low dimensionality task. Showing it would require demonstrating some more rapid appearance of its influence in a high dimensionality block..... I think the authors have the data to address this and it would be extremely interesting to know if this occurs. But I think they are not showing this. If they could either clarify what they mean or remove this term, I’d appreciate it.”*

Response: We agree with the Reviewer that the use of “latent” here is ambiguous. We were, in fact, using terminology from prior reports showing related findings. In this prior work, the notion of a latent representation is one that is dormant or hidden until the appropriate context or situation makes it apparent. Applied to our findings, we are referring to the fact that task-specific representations emerge throughout learning and that the nature of these representations vary

depending on the learning demands on the problems. However, latent is certainly an ambiguous term. Following the Reviewer's recommendation, we have removed "latent" from the discussion.

R2.6: *"Here are two papers that came to mind when reading that are not cited. Possibly they are not relevant to the authors, but just in case:"*

Response: We thank the Reviewer for these helpful references. They are now included in the revision.

Reviewer 3

Reviewer 3 thought our experimental design was clever and appreciated our use of a computational model to make inferences about more complex cognitive mechanisms. However, the Reviewer also had concerns with circularity in our analytic approach and that we should temper our claims.

R3.1: *"(1) If I understand the model correctly, it is based on the complexity of the BOLD response to the model that considers the multiple trials, with maximal complexity (dimensionality) being the case where there are responses to each of the trials that are different from one another. This would require a large set of dimensions to account for the variance in the BOLD signal patterns. Regions that could have 'compression' are necessarily those with minimal variable responses (minimal paradigm-locked response or constant response could both yield low dimensionality).*

Here is where I start to loose track of the relation between the model and the data. The sought property is an interaction of the dimensionality (as operationalized by the number of principal components that capture 90% of the variance) between the learning condition and learning block. This is a challenging model and unit to wrap ones head around, not simply because of its complexity but because of some ambiguities. For example, do high noise regions always have high PCA numbers because the 90% threshold is not achieved? Do regions with low response also always bend towards high dimensionality? Given change in dimensionality is the quantity being captured in compression, what is the actual change in signal? Does a robust highly variable signal converge to a more constant but clear response? Does the response modulation just go away? Is it diverse then becomes stationary across trials? Put simply, the operational definition of 'compression' on p. 5 leaves a great deal of the nuances of the data beneath the surface."

Response: We thank the Reviewer for noting the ambiguities in our original manuscript. We have revised the manuscript throughout but especially in the methods section (pg. 17) to clarify the steps of the PCA method and the factors that could impact the measure of neural compression. We have also added a new analysis that speaks to the nature of the representations underlying the change in dimensionality that we see across learning (Figure 4). Here, we will respond to each of these points in turn.

It is worth clarifying that the PCA method we propose depends on patterns of activation across voxels. Specifically, the first step is to transform the BOLD signal within each voxel for each trial into a summary measure of the fit to a canonical HRF. To do this, we leveraged the LS-S method (Mumford et al., 2012), a common approach for estimating single-trial activation

patterns relative to the task event structure across the rest of the dataset. PCA was then applied to the voxel patterns in these single-trial parameter estimates through a searchlight procedure. In the revision, we have clarified that PCA is conducted on patterns of activation as estimated with a single-trial GLM method rather than the amplitude of bold response (pg. 5). This approach does come with a limitation. Namely, single-trial GLMs are potentially impacted by signal to noise. Brain regions with lower signal to noise, either due to high levels of noise or low levels of signal, may lead to noisier parameter estimates. This noise in the GLM estimation could potentially influence the voxel patterns and lead to higher estimates of dimensionality from PCA. In the current study, since all comparisons were based on within-subject factors tested within the same brain region, this limitation would at worst lead to a false negative. In other words, the neural compression finding we see in vmPFC may in fact be present in other brain regions that have lower signal or higher noise, but the single trial GLM may have masked that effect. Of course, the limitation noted here is true of any analysis that depends on single-trial GLM estimates, including many forms of representational similarity analysis and betaseries connectivity methods. We were careful to only consider comparisons for within-subject factors thereby requiring a relative difference in neural compression across learning blocks and problem complexity. In the revised manuscript, we have added a warning to the discussion that describes the limitations due to single-trial GLM estimates (pg. 11) and expanded the methods description for neural compression to highlight the impact of signal to noise on PCA (pg. 17).

The Reviewer also raises an important question as to the nature of the neural signal that leads to the change in dimensionality that we observe. This question was also raised by the other Reviewers. In response to this question, we have included a new analysis that quantifies the degree of category discrimination present in the PCA results (Figure 4). Specifically, we examine how trials from the two categories load onto the PCs finding that trials from the same category load similarly and that the two categories have distinct loadings. This category discrimination in PC loadings emerges over learning and follows problem complexity such that low complexity has higher category discrimination. It is true that other factors could lead to lower dimensionality and these must be considered and accounted for when implemented the compression approach we propose. For example, a brain region may respond early in learning and decrease in activity as the task is mastered. Such a situation could be accompanied by an increase in compression across learning blocks. Two aspects of our analysis suggest this is not driving our findings. First, we included learning block accuracy in the mixed effects regression conducted in the searchlight analysis to account for compression changes due simply to changing accuracy. Second, a control univariate analysis did not show a significant interaction of learning block and problem complexity in BOLD activation within the vmPFC cluster (pg. 19). Thus, neural compression that we observed in vmPFC does not seem to arise from simple changes in BOLD signal, rather the problem-specific dimensionality reduction across learning seems to reflect organization of information according to category structure.

Again, we thank the Reviewer for noting the ambiguity in our description of the PCA method. We believe that the revised manuscript more appropriately describes the approach, notes the limitations, and offers a more nuanced view of how dimensionality in neural patterns can be impacted by a variety of factors, all of which need to be considered when indexing neural compression.

R3.2: *“(2) In a few places I worried that there was room for fishing and circularity in the estimated parameters and their certainty. In the methods, on p. 16, it is appropriately noted that the clusters were defined by the model and that post hoc analysis does not represent independent findings. However, the display of certainty and confidence intervals throughout,*

much like displaying biased r correlation values or inflated p values in traditional circular analyses, may be problematic. Take the '95% confidence bands' in the shaded plot within Figure 2B. Isn't that confidence interval derived from the idiosyncratic region defined in this sample that has been selected from a large number of possible regions? As such, the confidence bands, much like circularly defined p values, are almost certainly not an unbiased population estimate. Is this a problem for all of the data and scatter and confidence bands reported throughout the paper? Is the p value (10^{-5}) reported on p. 6 for the interaction derived from a biased selection of the region of 653 voxels chosen to have the tested property? I am unsure if there is or if there is not an issue in all cases, but the circular procedures (define map with a large number of mapped voxels, then select region representing a subset of voxels, then quantify behavior in that selected subset) smack of the kind of approach that has gotten the field into trouble in the past."

Response: The Reviewer is correct to worry about circularity as a general concern. We have been careful to explicitly acknowledge in the text that the vmPFC neural compression results reported in the plot of Figure 2B are not independent from the whole brain analysis (as the Reviewer notes) and are provided to demonstrate the nature of the learning block by problem complexity interaction. We have revised the legend for Figure 2 to also include this point. We have also changed our depiction of the results from the vmPFC cluster in Figure 2B. We now plot only the group means and participant-level data for neural compression from the peak voxel of the cluster. No depiction of error bars or confidence intervals are shown. We also have replaced the statistical tests on the average neural compression within the cluster in the main text with a statistical analysis of cluster compression from only the peak voxel of the cluster (pg. 6). This best represents the findings from the cluster as defined by the whole-brain mixed effects analysis.

To be clear, the analyses from both the original manuscript and the revised manuscript are free from circularity. The attention compression changes observed across learning and problem complexity are based only on model fits to behavioural responses. Thus, the analysis linking attention compression to vmPFC compression depends on the relationship between two independent data sources (behaviour and brain data). The newly-included category discrimination analysis (Figure 4) depends on a derived measure of PC loadings from the neural compression analysis and compares the differences in loadings for trials from different categories. PCA was conducted with no regard for the category label of the trials, thus a contrast of the PC loadings between categories is also free from circularity.

More generally, to provide a more rigorous statistical approach, we have opted to use Bayesian linear mixed effect models in our revision. Such an approach is less susceptible to over fitting, appropriately handles outliers, and provides probability distributions of model coefficients based on the data. We think this approach offers a clear view of our data and analysis, and helps to more accurately characterize the strength of our effects.

R3.3: *"(3) Given all of the assumptions of the model and the correlational nature of the data, the claims are too strong. The strong claim in the title that MPFC is active in compressing the concept representation is one such example."*

Response: We appreciate the Reviewer's check on our claims. In response to the other two Reviewers, we have extended our modelling approach to investigate the link between model and neural compression across the time course of learning. The positive results of this analysis provide stronger evidence in favour of vmPFC compression occurring as result of learning

processes. Additionally, the category discrimination present in the PC loadings (Figure 4) suggests that vmPFC representations do reflect the task structure of each problem in a manner consistent with dimensionality reduction. However, as the Reviewer aptly notes, we were too direct in the claims of our initial submission. As such, we have tempered the claims throughout, including a change to the title. These changes provide a better interpretation of our findings and make clear what open questions remain for future study.

Reviewers' Comments:

Reviewer #1:

Remarks to the Author:

The authors were thorough in addressing my feedback, in particular by including outcome regressors in the fMRI analysis, fitting the SUSTAIN model trial by trial, and unpacking its logic more. The main contribution of this paper is in introducing a novel index of neural compression which is an interesting measure to look at when evaluating models of representation learning. I do have some residual minor comments:

My previous comment regarding a strong claim about SUSTAIN resulted in relatively minor changes of phrasing, which do not fundamentally change the fact that the measure of neural compression based on PC number is on a much coarser scale than the model fitting of SUSTAIN to behavioral data. It would be good to acknowledge this explicitly.

Clarity of exposition can still be improved: what are the crosshairs in Fig 1C?

Please define all acronyms on first use (what is LS-S in the first paragraph of results?)

Fig 2a: I am confused about the compression score including trial number (n?) in it. Surely the n in the compression score is not the n you mention for enumerating trials.

In response to 1.1 the authors write that they included "regressors for feedback events (separately for correct and incorrect feedback)". From the methods it seems that they actually included a separate regressor for the outcome period for each trial. Which is correct, separate regressors for correct and incorrect feedback, or trial-specific regressors for outcome period?

R1.2: please note that there are still some instances of mPFC rather than vmPFC (e.g., in intro)

line 166: "suggesting a link between the behavioral and neural signatures of dimensionality reduction" -- I found this confusing -- the attention compression is as per the model, not a neural measure, and you are showing that the model compression (as fit to behavioral data, presumably) indeed correlates with the behavioral choices. I am not sure how this relates to neural signatures of dimensionality reduction in this particular analysis.

In general, it would be good to make clear for the different quantities described in the results (e.g., figure 3 caption), which is model-based, which is behavioral and which is neural. Since the different measures have very similar names, it is not clear and may be misleading.

I was not sure what is Bayesian linear regression (as opposed to regular linear regression)

line 248 -- did you mean borne? I'm not sure that fits in well, but born is somewhat strange in this context.

line 267 -- your description of reference 7 is somewhat inaccurate as the 16 states there were not goal states. They were just the different states of the task, which participants navigated. Notably, also, in that paper, the brain area discussed was the orbitofrontal cortex. It would help the reader, in my opinion, if somewhere early on you clarified the relationship between the area you are discussing and the orbitofrontal cortex. Are these one and the same? Conversely, what is the relationship between the area that you discuss and the vmPFC area identified in neuroeconomics studies of value and attention that you also mention? I don't believe these are all one and the same.

line 320 -- I feel that this statement is quite speculative and not necessarily warranted. The bar of

correlating significantly with a brain signal is quite a low one. For instance, signals in vmPFC correlate significantly with the prediction error from reinforcement learning models although it seems upon closer scrutiny that the signal in vmPFC is not a prediction error signal. Thus correlation between the brain signal and the model can't be taken as strong evidence for either the provenance of the brain signal or the correctness of the model.

Reviewer #2:

Remarks to the Author:

Thanks for addressing my concerns - responses and revisions look great.

Reviewer #3:

Remarks to the Author:

The authors have addressed my reviews but the complexity of the model and the difficulty in interpretation remain. In addition, the response acknowledges there are aspects of the process that may lead to some circularity (and hence overfitting).

Reviewer 1

R1.1: *“My previous comment regarding a strong claim about SUSTAIN resulted in relatively minor changes of phrasing, which do not fundamentally change the fact that the measure of neural compression based on PC number is on a much coarser scale than the model fitting of SUSTAIN to behavioral data. It would be good to acknowledge this explicitly.”*

Response: We thank the Reviewer for concisely noting this difference in scale between our neural measure of compression and the computational model-based analysis of behavioural. Indeed, the trial-by-trial model fitting we included in the last revision following Reviewer 1’s suggestion makes this difference all the more apparent. We have explicitly acknowledged the difference in scale between the measures (pg. 12).

R1.2: *“Clarity of exposition can still be improved: what are the crosshairs in Fig 1C? Please define all acronyms on first use (what is LS-S in the first paragraph of results?)”*

Response: Both of these issues have been fixed.

R1.3: *“Fig 2a: I am confused about the compression score including trial number (n?) in it. Surely the n in the compression score is not the n you mention for enumerating trials.”*

Response: The *n* in compression score does not refer to trial number, but the total number of trials in a learning block. We believe that this confusion was from the legend that referred to “each of *n* trial within a learning block” when it should have read “each of *n* trials within a learning block”. We have fixed this omission and thank the Reviewer for spotting it.

R1.4: *“In response to 1.1 the authors write that they included “regressors for feedback events (separately for correct and incorrect feedback)”. From the methods it seems that they actually included a separate regressor for the outcome period for each trial. Which is correct, separate regressors for correct and incorrect feedback, or trial-specific regressors for outcome period?”*

Response: We apologize for this confusion. Feedback events were modeled in two regressors for correct and incorrect feedback events. We have updated the methods text to appropriately reflect the final analysis reported in the manuscript (pg. 17).

R1.5: *“R1.2: please note that there are still some instances of mPFC rather than vmPFC (e.g., in intro)”*

Response: This has been fixed.

R1.6: *“line 166: ‘suggesting a link between the behavioral and neural signatures of dimensionality reduction’ -- I found this confusing -- the attention compression is as per the model, not a neural measure, and you are showing that the model compression (as fit to behavioral data, presumably) indeed correlates with the behavioral choices. I am not sure how this relates to neural signatures of dimensionality reduction in this particular analysis.”*

Response: The first half of this sentence describes that attention compression (based on computational model fits to behaviour) varies with problem complexity in a similar manner as to

how neural compression (which is computed with no regard to behavioural fits of the computational model) varies with problem complexity. This similarity is the basis for our suggestion that there may be a potential link behavioural and neural signatures of dimensionality reduction. We have rephrased this sentence to make the basis for our suggestion clear (pg. 7).

R1.7: *“In general, it would be good to make clear for the different quantities described in the results (e.g., figure 3 caption), which is model-based, which is behavioral and which is neural. Since the different measures have very similar names, it is not clear and may be misleading.”*

Response: We have clarified the basis for the different quantities throughout the manuscript, but especially in the Figure 3 caption, by noting whether a quantity is derived from neural or model-based measures.

R1.8: *“I was not sure what is Bayesian linear regression (as opposed to regular linear regression)”*

Response: In response to comments from other Reviewers, we reanalyzed our data with linear regression models based on Bayesian estimation with the rstanarm R package. The underlying statistical model is the same as a regular linear regression, however the parameters of the model are estimated through hierarchical Bayesian methods. Such an approach allows for a better characterization of the factors of interest in light the actual data without the constraints of normality and other distribution-specific characteristics. We have included details about the Bayesian estimation approach in the methods section (pg. 19). Also, we have relabeled references to these analyses as “Bayesian-estimated linear regression” throughout the manuscript to more correctly reflect the approach.

R1.9: *“line 248 -- did you mean borne? I'm not sure that fits in well, but born is somewhat strange in this context.”*

Response: We have rephrased this sentence.

R1.10: *“line 267 -- your description of reference 7 is somewhat inaccurate as the 16 states there were not goal states. They were just the different states of the task, which participants navigated. Notably, also, in that paper, the brain area discussed was the orbitofrontal cortex. It would help the reader, in my opinion, if somewhere early on you clarified the relationship between the area you are discussing and the orbitofrontal cortex. Are these one and the same?”*

Response: We thank the Reviewer for noting this inaccurate description, we have updated this to refer to task rather than goal states. Also, we have included a discussion of the relationship between OFC and vmPFC noting the fuzzy boundaries associated with vmPFC due to functional and anatomical divisions (pg. 10) to help bridge the different labels used across the literature.

R1.11: *“Conversely, what is the relationship between the area that you discuss and the vmPFC area identified in neuroeconomics studies of value and attention that you also mention? I don't believe these are all one and the same.”*

Response: The review paper and study on value representations that we reference actually find evidence for such value representations in regions that overlap with the vmPFC area we identified. It was, in fact, this overlap that led to our suggestion that dimensionality reduction may provide a domain-general mechanism for explaining value and attention effects in vmPFC.

R1.12: *“line 320 -- I feel that this statement is quite speculative and not necessarily warranted. The bar of correlating significantly with a brain signal is quite a low one. For instance, signals in vmPFC correlate significantly with the prediction error from reinforcement learning models although it seems upon closer scrutiny that the signal in vmPFC is not a prediction error signal. Thus correlation between the brain signal and the model can't be taken as strong evidence for either the provenance of the brain signal or the correctness of the model.”*

Response: The intended meaning of this sentence was that a competing model would have to provide an account of attention/representation compression that is related to vmPFC neural compression at least to the same degree as we observe with SUSTAIN in the current work. Thus, the “high bar” was in relation to a comparison between models (i.e., any model has to fit as well or better than SUSTAIN). This is perhaps a subtle point and one not necessary to make in this discussion. As such, we have removed this sentence (pg. 11).

Reviewer 2

“Thanks for addressing my concerns - responses and revisions look great.”

We greatly appreciate Reviewer 2's constructive suggestions and enthusiasm for our work throughout the review process.

Reviewer 3

“The authors have addressed my reviews but the complexity of the model and the difficulty in interpretation remain. In addition, the response acknowledges there are aspects of the process that may lead to some circularity (and hence overfitting).”

Response: We have substantially revised our discussion to include further points about factors that may influence neural compression and the limitations such factors pose for the current study. We have also provided further discussion on the question of circularity (pg. 12-13).